Corrected: Author correction

# Iron restriction inside macrophages regulates pulmonary host defense against *Rhizopus* species

Angeliki M. Andrianaki[1], Irene Kyrmizi[1,2], Kalliopi Thanopoulou[3], Clara Baldin[4], Elias Drakos[1], Sameh S.M. Soliman[5], Amol C. Shetty[6], Carrie McCracken[6], Tonia Akoumianaki[1], Kostas Stylianou[1], Petros Ioannou [1], Charalampos Pontikoglou[1], Helen A. Papadaki[1], Maria Tzardi[1], Valerie Belle[7], Emilien Etienne[7], Anne Beauvais[8], George Samonis[1], Dimitrios P. Kontoyiannis[9], Evangelos Andreakos[3], Vincent M. Bruno[6], Ashraf S. Ibrahim[4,10] & Georgios Chamilos [1,2]

Mucormycosis is a life-threatening respiratory fungal infection predominantly caused by *Rhizopus* species. Mucormycosis has incompletely understood pathogenesis, particularly how abnormalities in iron metabolism compromise immune responses. Here we show how, as opposed to other filamentous fungi, *Rhizopus* spp. establish intracellular persistence inside alveolar macrophages (AMs). Mechanistically, lack of intracellular swelling of *Rhizopus* conidia results in surface retention of melanin, which induces phagosome maturation arrest through inhibition of LC3-associated phagocytosis. Intracellular inhibition of *Rhizopus* is an important effector mechanism, as infection of immunocompetent mice with swollen conidia, which evade phagocytosis, results in acute lethality. Concordantly, AM depletion markedly increases susceptibility to mucormycosis. Host and pathogen transcriptomics, iron supplementation studies, and genetic manipulation of iron assimilation of fungal pathways demonstrate that iron restriction inside macrophages regulates immunity against *Rhizopus*. Our findings shed light on the pathogenetic mechanisms of mucormycosis and reveal the role of macrophage-mediated nutritional immunity against filamentous fungi.

[1] Department of Medicine, University of Crete, Foundation for Research and Technology, 71300 Heraklion, Crete, Greece. [2] Institute of Molecular Biology and Biotechnology, Foundation for Research and Technology, 71300 Heraklion, Crete, Greece. [3] Laboratory of Immunobiology, Center for Clinical, Experimental Surgery, and Translational Research, Biomedical Research Foundation of the Academy of Athens, 11527 Athens, Greece. [4] Division of Infectious Diseases, Los Angeles Biomedical Research Institute, Harbor-University of California Los Angeles (UCLA) Medical Center, 1124 West Carson Street, St. John's Cardiovascular Research Center, Torrance, CA 90502, USA. [5] Sharjah Institute for Medical Research, College of Pharmacy, University of Sharjah, PO Box 27272 Sharjah, UAE. [6] Institute for Genome Sciences, University of Maryland School of Medicine, Baltimore, MD 21201, USA. [7] CNRS, BIP (UMR 7281), IMM (FR 3479), Aix-Marseille Université, 31 chemin J. Aiguier, 13402 Marseille, France. [8] Unité des Aspergillus, Institut Pasteur, 75015 Paris, France. [9] Department of Infectious Diseases, Infection Control, and Employee Health, The University of Texas MD Anderson Cancer Center, Houston, TX 77030, USA. [10] David Geffen School of Medicine at UCLA, Los Angeles, CA 90095, USA. These authors contributed equally: Angeliki M. Andrianaki, Irene Kyrmizi. Correspondence and requests for materials should be addressed to A.S.I. (email: ibrahim@labiomed.org) or to G.C. (email: hamilos@imbb.forth.gr)

Respiratory fungal infections caused by opportunistic filamentous fungi, predominantly *Aspergillus* species and Mucorales, are major causes of death in an expanding population of debilitated and immunocompromised patients[1–3]. When compared to other airborne filamentous fungi (molds), Mucorales are emerging pathogens that possess unique clinical, epidemiological, and pathogenetic characteristics[4–8]. Specifically, although acquired innate immune defects associated with myeloablative and immunosuppressive therapies are typical risk factors for mucormycosis, Mucorales frequently cause lethal infections in patients with certain metabolic disorders, and occasionally in immunocompetent individuals following trauma and burns[6–8]. In particular, patients with diabetic ketoacidosis or other forms of acidosis, and patients with elevated available serum iron are uniquely predisposed to mucormycosis[6–8]. Importantly, pulmonary mucormycosis is the most devastating invasive fungal infection, associated with mortality rates of 50% to 70% and can reach 90% upon dissemination[6–8]. This disease also has significant morbidity and frequently requires disfiguring radical surgery as the only lifesaving measure[6–8]. Because of the inherent resistance of Mucorales to most available antifungal agents[8], there is a dire need to better understand immunopathogenesis of mucormycosis and develop new therapeutic concepts for the disease.

Iron has an essential role in the life cycle of Mucorales and its utilization from the host is a critical pathogenetic mechanism of mucormycosis[7,8]. In contrast, the host-mediated iron limitation is an important defense strategy known as nutritional immunity. Recent genetic and biochemical studies in *Rhizopus*, the Mucorales causing 70% of human mucormycosis, have identified key molecular determinants of iron assimilation that are required for invasive fungal growth. Specifically, studies highlighted the role of the high affinity iron permease (Ftr1p) in iron acquisition from iron-limited environment encountered by the fungus during in vivo infection[9]. Additionally, the fungal proteins Fob1 and Fob2 act specifically as receptors for iron uptake from ferrioxamine[10]. Ferrioxamine is the iron-rich form of deferoxamine (DFO), which is utilized by the fungus as a xenosiderophore. The use of DFO for iron chelation therapy in dialysis patients with end-stage renal failure is associated with unique predisposition to mucormycosis[11,12]. Furthermore, in patients with diabetic ketoacidosis or other forms of acidosis, increased availability of free iron in the serum as a result of protonation of transferrin, is a critical pathogenetic event in the development of mucormycosis[8]. Notably, correction of diabetic acidosis with bicarbonate has been an effective adjunct treatment of mucormycosis in mice[13]. Also, abnormalities of iron metabolism and hyperglycemia facilitate interaction of CotH3 Mucorales invasin[14] with endothelial GRP78 receptor[15] and induce angioinvasive fungal growth, another hallmark of mucormycosis.

Despite the progress in identifying important iron assimilation pathways in Mucorales, the host effector mechanisms that limit iron availability to these pathogens are incompletely characterized. Furthermore, there is no pathogenetic model to explain whether and how abnormalities in iron metabolism compromise physiological immune responses leading to development of mucormycosis.

It is known that professional phagocytes, including neutrophils and macrophages, have a central role in host defense against airborne filamentous fungi[1,16]. However, there is a significant gap of knowledge on physiological immune responses against Mucorales. In fact, current evidence on the role of cellular immunity against Mucorales is extrapolated from studies on *Aspergillus*[16]. Similarly, at the molecular level, little is known about the mechanisms of intracellular killing of Mucorales

conidia inside phagocytes. From the pathogen perspective, the presence of putative virulence mechanisms that protect Mucorales conidia from killing by phagocytes remain elusive.

Herein, we analyze physiological immune responses in the lungs of immunocompetent mice during *Rhizopus* infection and identify an essential role of macrophages during host–*Rhizopus* interplay. In particular, we demonstrate that *Rhizopus* spp. subvert physiological killing mechanisms of macrophages and establish prolonged intracellular dormancy via melanin-induced phagosome maturation arrest. Furthermore, we discover that inhibition of *Rhizopus* growth inside macrophages is a central host defense mechanism that depends on nutritional immunity via iron starvation. Finally, dual RNA-sequencing (RNA-seq) and functional studies identify critical host and fungal modulators of iron homeostasis inside macrophages that promote invasive fungal growth. Our findings shed light on pathogenesis of mucormycosis with potential therapeutic implications in the management of this devastating disease, and provide a mechanistic link between iron homeostasis inside macrophages and immune responses to Mucorales.

## Results

**Rhizopus conidia persist inside alveolar macrophages.** In order to understand general aspects of the physiological immune response against Mucorales, we infected immunocompetent mice with conidia of either *Aspergillus fumigatus* or two clinical isolates of *Rhizopus* (*Rhizopus oryzae* and *Rhizopus delemar*) and compared fungal clearance by CFU (colony-forming unit) counts and histopathological studies of total lung homogenates at different time points of infection (0, 2, 5, and 10 days). We found that opposite to *A. fumigatus* conidia, a significant proportion of each *Rhizopus* isolate conidia remained viable in the lungs of immunocompetent mice as late as 10 days post infection (Fig. 1a), despite the lack of mortality of the infected animals (Fig. 1b). Histopathology of lung tissue sections collected on day 5 post infecting immunocompetent mice with *R. oryzae* confirmed the presence of abundant conidia in the lungs, which resulted in considerable tissue edema and neutrophil infiltration (Fig. 1c). Interestingly, there was no apparent evidence of *R. oryzae* germination in infected lungs (Fig. 1c). Furthermore, immunohistochemistry studies revealed that *R. oryzae* conidia predominantly resided inside CD11c+/CD68+cells consistent with alveolar macrophages (AMs), while there was evidence of extracellular conidia associated with areas of intensive neutrophil infiltration (Fig. 1d).

We next assessed kinetics of recruitment and degree of association of myeloid cells in the lungs of immunocompetent mice following infection with fluorescence-labeled conidia of *R. oryzae*. Although we found a significant influx of neutrophils (identified as CD45+/CD11b+/Ly6G+/Ly6C− cells; Supplementary Fig. 1) and Ly6C$^{high}$ inflammatory monocytes (identified as CD45+/CD11b+/Ly6G−/CD11c−/Ly6C+ cells; Supplementary Fig. 1) in the lungs on days 1 and 5 of infection (Fig. 1e), most *R. oryzae* conidia were associated with AMs (identified as CD45+/F4/80+/CD11c+/MHCII$^{low}$ cells), interstitial macrophages (IMs; identified as CD45+/F4/80+/CD11c−/MHCII+), and dendritic cells (DCs; identified as CD45+/F4/80−/CD11c+/MHCII+) in the lungs of immunocompetent mice (Fig. 1f). Confocal imaging in sorted neutrophils, AMs, and IMs on day 5 post infection (Supplementary Fig. 2) confirmed that *R. oryzae* conidia were predominantly phagocytosed by AMs (Fig. 1g, h). Collectively, these studies demonstrate that a significant proportion of *Rhizopus* conidia remain viable inside AMs for several days following infection of immunocompetent mice.

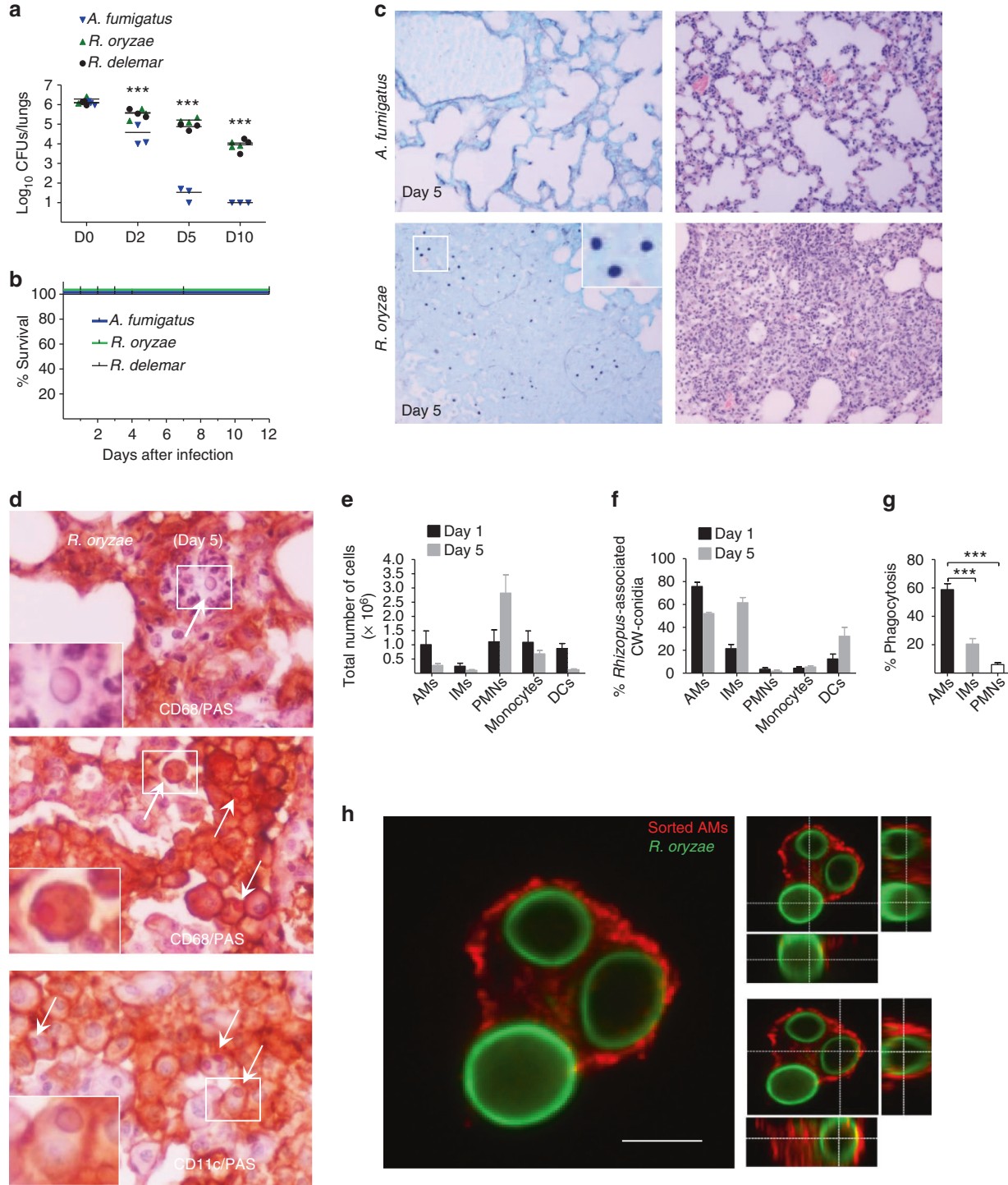

**Rhizopus conidia are resistant to killing by macrophages**. In order to shed light on the mechanisms of intracellular persistence of *Rhizopus*, we studied interactions of bone marrow-derived macrophages (BMDMs) and neutrophils with conidia of *A. fumigatus* or *R. oryzae*. In contrast to conidia of *A. fumigatus*, which were phagocytosed comparably by BMDMs and neutrophils, conidia of *R. oryzae* were almost exclusively phagocytosed by BMDMs (Fig. 2a–c). In addition, despite their larger size, *Rhizopus* conidia were phagocytosed by BMDMs at a higher index rate than *A. fumigatus* conidia. Next, we compared killing rates of *A. fumigatus* vs. *R. oryzae* conidia by BMDMs. We found

that unlike *A. fumigatus* (Fig. 2d, Supplementary Fig. 3), different clinical isolates of *Rhizopus* were resistant to killing by BMDMs (Fig. 2d, e, Supplementary Fig. 3). These studies corroborate the results of in vivo studies on selective tropism and prolonged intracellular persistence of *Rhizopus* conidia inside AMs.

In order to evaluate the mechanism of *Rhizopus* persistence inside macrophages, we tested whether fungal conidia are resistant to damage by phagocyte effector mechanisms[4,7,8]. Therefore, we compared susceptibility of *A. fumigatus* vs. *R. oryzae* conidia to (i) oxidative damage induced by hydrogen peroxide ($H_2O_2$) and (ii) non-oxidative damage mediated by

**Fig. 1** Persistence of *Rhizopus* conidia inside alveolar macrophages (AMs). **a** Fungal loads in lungs of immunocompetent C57BL/6 (B6) mice ($n = 3$ per group) infected via intratracheal administration of a standardized inoculum ($5 \times 10^6$ conidia per mice) of *A. fumigatus*, *R. oryzae*, or *R. delemar*. ***$P < 0.0001$, Mann–Whitney test. **b** Survival of immunocompetent C57BL/6 (B6) mice ($n = 8$ per group) infected as in **a**, with either *A. fumigatus*, *R. oryzae*, or *R. delemar*. **c** Representative photomicrographs of the lungs from mice infected as in **b** with either *A. fumigatus* or *R. oryzae* and sacrificed on day 5. Histopathological sections were stained with Grocott methenamine silver (GMS; left panels) or hematoxylin and eosin (H&E; right panels). The presence of *R. oryzae* conidia (black color) in the lungs is shown by GME stain. Original magnification ×400. **d** Representative photomicrographs of the lungs from mice infected as in **b**, sacrificed on day 5. Lungs were stained by IHC for CD68 or CD11 and counterstained with hematoxylin and PAS. There is evidence of extracellular *R. oryzae* conidia surrounded by neutrophils (top panel), and intracellular *R. oryzae* conidia inside AMs. **e** FACS analysis of total number of professional phagocytes in the lungs of immunocompetent mice ($n = 3$ per group) infected with *R. oryzae* as in **a**, assessed on days 1 and 5. The gating strategy for identification of neutrophils, monocytes, AMs, interstitial macrophages (IMs), and dendritic cells (DCs) is shown in Supplementary Fig. 1. **f** FACS analysis of association of labeled (Fluorescent Brightener 28; CW) conidia of *R. oryzae* with professional phagocytes of mice infected as in **a**, assessed on days 1 and 5. **g** In vivo phagocytosis rates of *R. oryzae* conidia on day 5 of infection of mice infected as in **a**. ***$P < 0.0001$ Mann–Whitney test. **h** Representative confocal image of sorted AM from mice infected as in **a** with fluorescent-labeled, live conidia of *R. oryzae* (day 1), fixed and stained with Cathepsin D. Cross-section analysis was performed to discriminate intracellular conidia from conidia associated/bound to the cell surface of AM

lysosomal hydrolases. Importantly, conidia of both fungi displayed comparable degree of susceptibility to killing by increasing concentrations of either $H_2O_2$ (Fig. 2f) or crude lysosomal extracts (Fig. 2g, h). Therefore, both fungi displayed comparable degree of susceptibility to macrophage effector killing mechanisms and the inability of BMDMs to kill *Rhizopus* could not be explained by resistance of the conidia to oxidative or non-oxidative damage.

During interaction of human fungal pathogens with macrophages, the induction of different forms of host cell death is a survival strategy that allows the fungus to escape host defense[17–19]. In addition, studies on *Mucor circinelloides* demonstrated that fungal sporangiospores germinate and induce killing of a murine macrophage cell line[20]. Therefore, we investigated if *A. fumigatus* and *R. oryzae* vary in their ability to induce host cell apoptosis or necrosis. We found no difference between the two fungi in induction of either form of host cell death during interaction with murine BMDMs (Fig. 2i). Collectively, these data demonstrate that Mucorales survive inside macrophages by avoiding phagocyte-mediated killing.

**Conidia induce phagosome maturation arrest via targeting LAP.** To understand the mechanism of establishment of *Rhizopus* intracellular persistence, we performed comparative phagosome biogenesis studies following phagocytosis of *A. fumigatus* or *R. oryzae* by BMDMs. Because LC3-associated phagocytosis (LAP) is a major antifungal pathway regulating early events in biogenesis of *A. fumigatus* phagosome[21,22], we initially analyzed LC3+ phagosome (LAPosome) formation at different time points of infection by confocal imaging. In sharp contrast to the robust activation of LAP pathway upon infection with *A. fumigatus*, we found no evidence of LAPosome formation during infection of BMDMs with *R. oryzae* (Fig. 3a, b). Similarly, Rab5 recruitment to the phagosome, an early event in phagosome maturation process[23], selectively and transiently occurred at early time points of BMDM infection with *A. fumigatus*, while there was no evidence of Rab5 localization in *R. oryzae*-containing phagosomes (Fig. 3c, Supplementary Fig. 4).

Next, we monitored phagolysosomal (P–L) fusion upon BMDM infection with *A. fumigatus* vs. *R. oryzae* with the use of fluorescein isothiocyanate-labeled dextran (FITC-Dextran) assay. Briefly, BMDMs were pre-loaded with FITC-Dextran for labeling of lysosomes and P–L fusion upon fungal infection was assessed based on accumulation of FITC-Dextran in the phagosomes. In contrast to the accumulation of FITC-Dextran in *A. fumigatus* containing phagosomes over time (Fig. 3d, e), there was no evidence of FITC-Dextran recruitment in Mucorales phagosomes. Concordant with these results, recruitment of Cathepsin D, a lysosomal hydrolase, in phagosomes was

significantly impaired at 4 h of BMDM infection with *R. oryzae* as compared to *A. fumigatus* (Fig. 3f, g). Further, these findings were confirmed by electron microscopy studies on fusion of acid phosphatase-stained lysosomes with *A. fumigatus* vs. *R. oryzae* phagosomes. Notably, while P–L fusion was evidenced as intense acid phosphatase staining in *A. fumigatus* phagosomes at 4 h of infection, there was no lysosomal association with *R. oryzae* phagosomes (Fig. 3h). Similarly, in vivo studies in AMs of immunocompetent mice following infection with *R. oryzae* revealed the ability of Mucorales conidia to block P–L fusion by inhibiting LAP (Fig. 4a–d). Collectively, these studies confirm that *Rhizopus* conidia block LAP and induce phagosome maturation arrest to establish intracellular persistence inside the macrophage.

**Cell wall melanin induces phagosome maturation arrest.** We have recently demonstrated that cell wall melanin on dormant conidia of *A. fumigatus* blocks LAP to inhibit phagosome biogenesis and promote fungal pathogenicity[22]. In addition, we revealed that removal of melanin during intracellular swelling of *A. fumigatus* conidia is a fundamental requirement for activation of LAP and efficient killing of the fungus by monocytes/macrophages[21,22]. Thus, we assessed whether a similar mechanism is responsible for the phagosome maturation arrest seen with *R. oryzae* conidia. Intriguingly, in sharp contrast to *A. fumigatus* conidia, *Rhizopus* conidia remained dormant without evidence of cell wall swelling at 24 h of infection in macrophages ex vivo (Fig. 5a, b) and in vivo, inside AMs (Supplementary Fig. 5). This finding corroborates the lack of apparent germination of *R. oryzae* conidia in the lungs of immunocompetent mice. Therefore, in all likelihood, lack of cell wall remodeling in Mucorales conidia results in surface retention of cell wall inhibitory molecule(s) to block phagosome responses and establish intracellular persistence. Consistent with this hypothesis, we found that ultraviolet (UV)-inactivated conidia of Mucorales induce phagosome maturation arrest in macrophages to a similar degree of live conidia (Supplementary Fig. 6), implying the presence of an inhibitory molecule on the fungal cell wall surface.

Despite the differences in cell wall composition of filamentous fungi, melanin is a common cell wall molecule in conidia of many filamentous fungi including *Aspergillus*[24,25]. Therefore, we purified melanin from conidial cell wall of *Rhizopus* spp. and performed comparative characterization with *A. fumigatus* 1, 8-dihydroxynaphthalene (DHN) melanin. Notably, we found that *Rhizopus* melanin exhibited all the classic characteristics of melanin pigments (Supplementary Table 1). However, electron paramagnetic resonance (EPR) spectra (Fig. 5c), UV spectrophotometry (Supplementary Fig. 7), infrared (IR) spectroscopy (Supplementary Fig. 8), and chemical degradation followed by

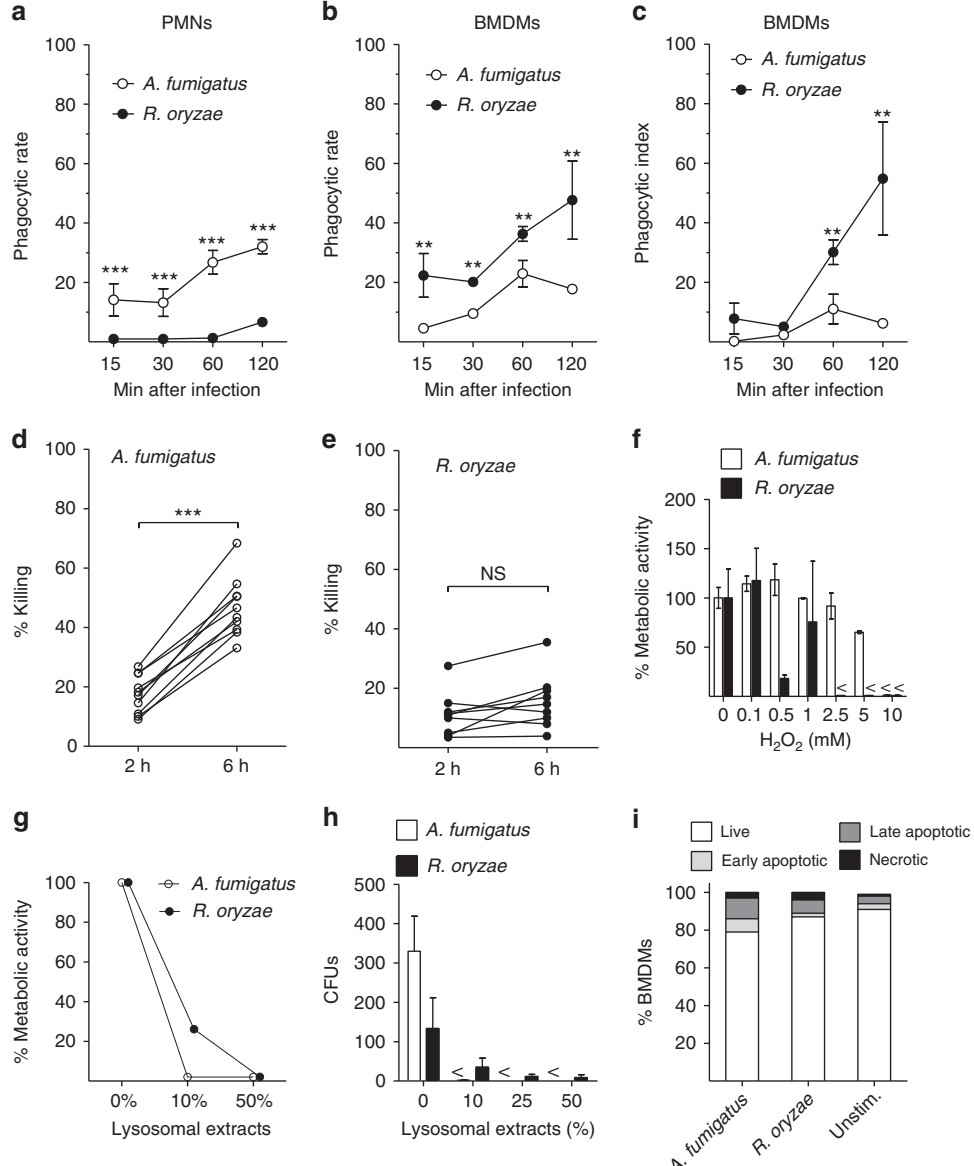

**Fig. 2** *Rhizopus* conidia display resistance to ex vivo killing by macrophages. **a**–**c** Comparative studies on phagocytosis of *A. fumigatus* and *R. oryzae* conidia by BMDMs and neutrophils (PMNs) assessed by confocal imaging. Data on quantification of phagocytosis are presented as mean ± SEM of five independent experiments ***$P < 0.0001$, **$P < 0.001$, Mann–Whitney test. **d**, **e** Intracellular killing of *A. fumigatus* and *R. oryzae* (Mucorales) conidia by BMDMs. Symbols connected with a line represent time points of the same independent experiment ($n = 9$ per group). ***$P < 0.0001$, Mann–Whitney test. **f**, **g** Assessment of in vitro susceptibility of *A. fumigatus* and *R. oryzae* conidia to **f** oxidative damage induced by increasing concentrations of $H_2O_2$ or **g** to damage induced by enzymatic activity of increasing concentrations of lysosomal extracts of BMDMs, assessed by measurement of fungal metabolic activity using the XTT assay at 24 h. **h** In vitro fungicidal activity of increasing concentrations of lysosomal extracts against conidia of *A. fumigatus* or *R. oryzae* assessed by CFU plating. **i** Induction of apoptosis in unstimulated BMDMs or BMDMs infected with *A. fumigatus* or *R. oryzae* at an MOI of 3:1 (effector:fungal cells) for 6 h. Apoptotic BMDMs were assessed by FACS analysis following Annexin V/PI staining. Data are representative of one out of three independent experiments. *NS* not significant

liquid chromatography-mass spectrometry (LC-MS) studies of the degradation products (Supplementary Fig. 9) revealed that *Rhizopus* melanin is different than DHN melanin of *A. fumigatus*. In particular, *Rhizopus* melanin has chemical composition consistent of eumelanin, and requires a biosynthetic pathway, which likely involves activity of a copper-dependent tyrosinase/laccase (Supplementary Figs. 7, 8 and 9). Indeed, growth in copper-free media resulted in the production of albino conidia of *Rhizopus*.

Importantly, functional studies on phagosome responses in macrophages stimulated with purified melanin of either

*Aspergillus* or *Rhizopus* demonstrated that both types of melanin induced complete phagosome maturation arrest (Fig. 5d). As a further proof of the melanin-dependent induction of phagosome maturation arrest by intracellular conidia of *Rhizopus*, we assessed LAPosome formation and Cathepsin D recruitment upon chemical degradation of *Rhizopus* melanin via $H_2O_2$ bleaching, as previously described[22]. Importantly, chemical removal of melanin from conidia of *Rhizopus* resulted in reversal of phagosome maturation arrest as evidenced by the significant increase in recruitment of LC3 II (Fig. 5e, f) and Cathepsin D (Fig. 5g, h) to the phagosomes containing melanin-deficient

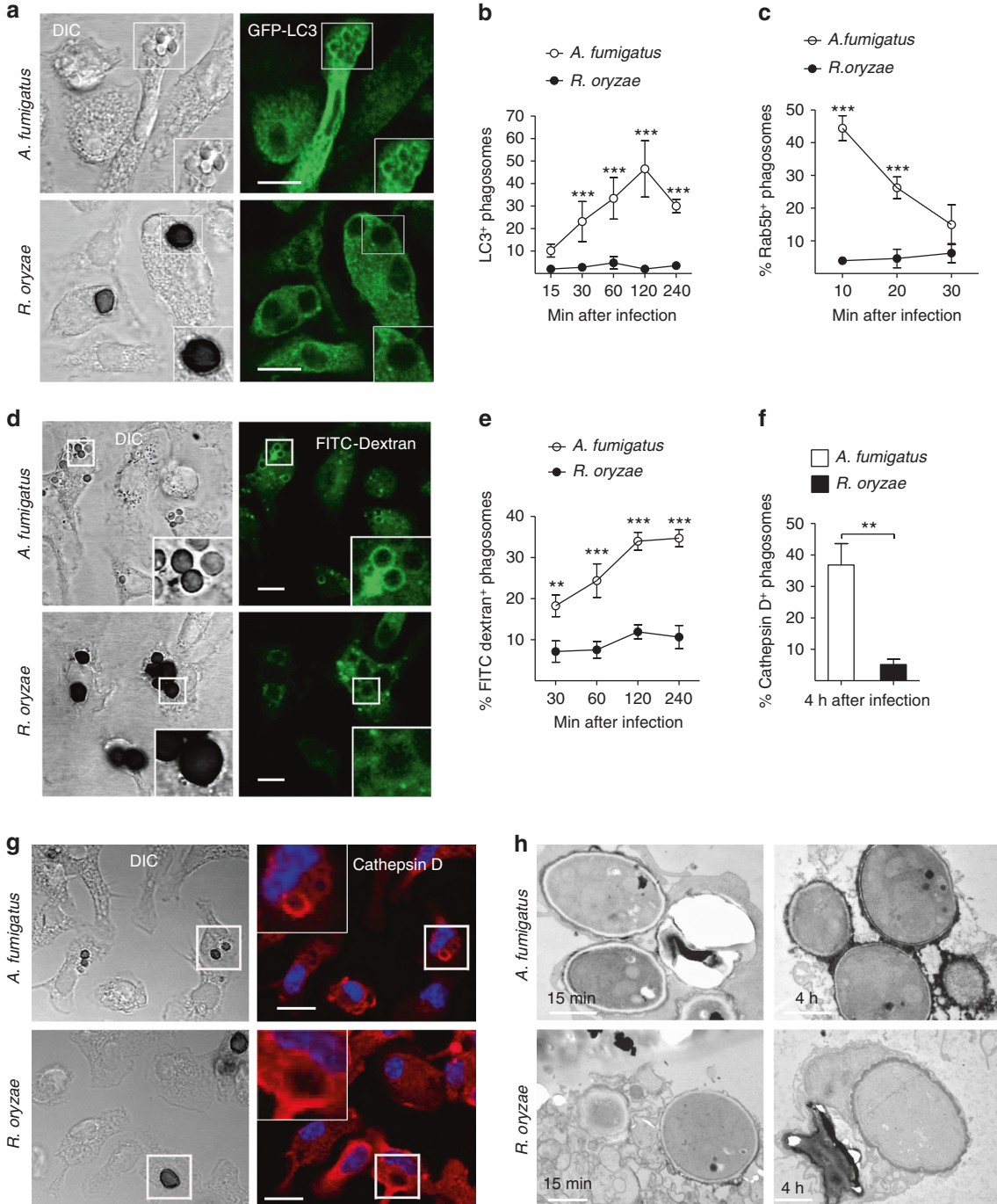

**Fig. 3** *Rhizopus* conidia induce phagosome maturation arrest in BMDMs. **a**, **b** BMDMs from GFP-LC3 mice were infected at different time points with live conidia of *A. fumigatus* or *R. oryzae* at an MOI 3:1 (effector:fungal cells). At the indicated time point, cells were fixed and analyzed by confocal microscopy. Representative fluorescence images are presented in **a**. Bar, 5 μm. Data on quantification of LC3+ phagosomes are presented as mean ± SEM of five independent experiments in **b**. **c** BMDMs from C57BL/6 (B6) mice were infected as in **a**. Cells were fixed at the indicated time point and stained for Rab5B. Data on quantification of Rab5+ phagosomes are presented as mean ± SEM of three independent experiments. **d**, **e** BMDMs were pre-loaded with FITC-Dextran, infected as in **a**, and phagolysosomal fusion was assessed at the indicated time point based on acquisition of FITC-Dextran in the phagosome. Representative fluorescence images are shown in **d**. Bar, 5 μm. Data on quantification of FITC-Dextran+ phagosomes are presented in **e** as mean ± SEM of three independent experiments. **f**, **g** BMDMs were stimulated as in **a**, cells were fixed and stained for the lysosomal protein marker Cathepsin D. Data on quantification of Cathepsin D+ phagosomes are presented as mean ± SEM of three independent experiments in **f**, while representative fluorescence images are shown in **g**. Bar, 5 μm. **h** Representative electron microscopy of acid phosphatase, a lysosomal enzyme marker of phagolysosomal fusion (shown as dark color on the phagosome membrane) in 15 min and 4 h phagosomes containing *A. fumigatus* or *R. oryzae* conidia. ***P < 0.0001, **P < 0.01, Mann–Whitney test. Bar, left upper and lower panels, 1 μm; right upper panel, 0.5 μm; right lower panel, 2 μm

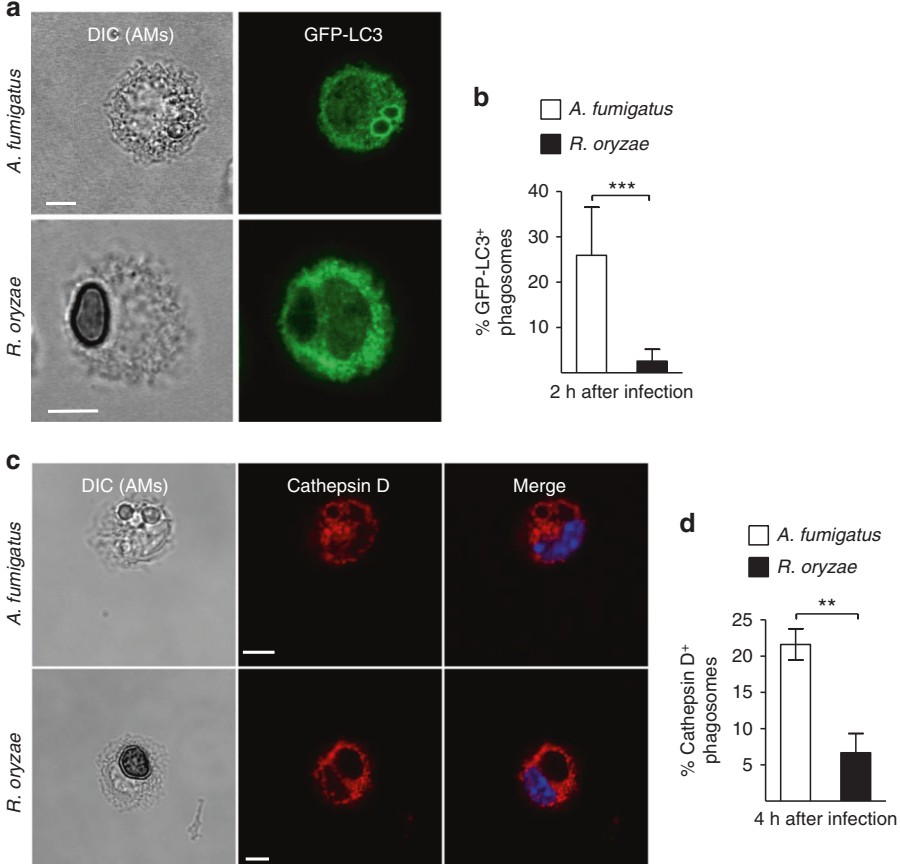

**Fig. 4** *Rhizopus* inhibits phagosome biogenesis in AMs during in vivo infection. GFP-LC3 (**a**, **b**) or C57BL/6 (B6) (**c**, **d**) mice were infected intratracheally with 5 × 10⁶ conidia of either *R. oryzae* or *A. fumigatus*. AMs were obtained by bronchoalveolar lavage at the indicated time point, fixed, stained with anti-GFP (**a**) or Cathepsin D (**b**) antibodies, and assessed by confocal imaging. Representative fluorescence images are shown (**a**, **c**). Data on quantification of GFP-LC3⁺ (**b**) and Cathepsin D⁺ (**d**) phagosomes are presented as mean ± SEM of three independent experiments. ***$P < 0.0001$, **$P < 0.001$ Mann–Whitney test. Scale bar, 5 μm

(albino) *Rhizopus*. Finally, infection of immunocompetent mice with melanin-deficient (albino) *Rhizopus* conidia recovered from culture in copper-free media (Fig. 5i) resulted in rapid fungal clearance from the lungs, when compared to infection with wild-type (WT) (control) melanized *Rhizopus* conidia (Fig. 5j). Collectively, these findings demonstrate that surface retention of melanin in dormant conidia of *Rhizopus* inside macrophages is the mechanism that accounts for prolonged intracellular persistence.

**Non-redundant role of macrophages in pulmonary mucormycosis.** Despite the inability of macrophages to kill *Rhizopus*, inhibition of growth of phagocytosed conidia could be an important host defense mechanism. To test this hypothesis, we initially assessed differences in phagocytosis of dormant vs. germinating (swollen) conidia following 4 h pre-incubation in culture media. Notably, we found a significant ≈5-fold reduction in phagocytosis of swollen as compared to dormant conidia by macrophages (Fig. 6a).

Then, we infected immunocompetent mice via intratracheal administration of the same inoculum of dormant vs. swollen conidia of *Rhizopus* (5 × 10⁶ conidia/mice), and assessed differences in the infection outcome. Surprisingly, while all mice recovered from infection with dormant *Rhizopus* conidia, infection of immunocompetent mice with swollen conidia of *Rhizopus* resulted in 100% mortality within 4 days (Fig. 6b).

Histopathological examination of the lungs at 16 h of infection with swollen *Rhizopus* conidia revealed dense neutrophil infiltrates along with extensive tissue necrosis and angioinvasive fungal growth (Fig. 6c, Supplementary Fig. 10). Notably, we found no evidence of *Rhizopus* inside AMs or other phagocytes following infection of mice with swollen conidia (Fig. 6c, Supplementary Fig. 10), a finding consistent with the results of ex vivo studies.

In order to directly evaluate the role of macrophages in physiological immune response to Mucorales, we performed AM depletion via intratracheal administration of clodronate liposomes[26,27] and 3 days latter infected mice with *Rhizopus* dormant conidia. Interestingly, survival, histopathology, and fungal load experiments demonstrate that liposome depletion resulted in significant increase in susceptibility of mice to mucormycosis as compared to treatment with control liposomes (Fig. 6d–f). In order to further validate the role of AMs in mucormycosis, we depleted CD11c transgenic mice with the administration of diphtheria toxin (DT), verified selective depletion of CD11c+ cells (Fig. 6g, Supplementary Fig. 11), and assessed the effect on survival following *Rhizopus* infection. Importantly, we found a dramatic increase in susceptibility of CD11c-depleted mice to mucormycosis with 100% mortality at 5 days of infection (Fig. 6h). Collectively, these findings demonstrate a non-redundant role of CD11c+ cells in the lungs, including AMs and DCs, in antifungal immunity against *Rhizopus*.

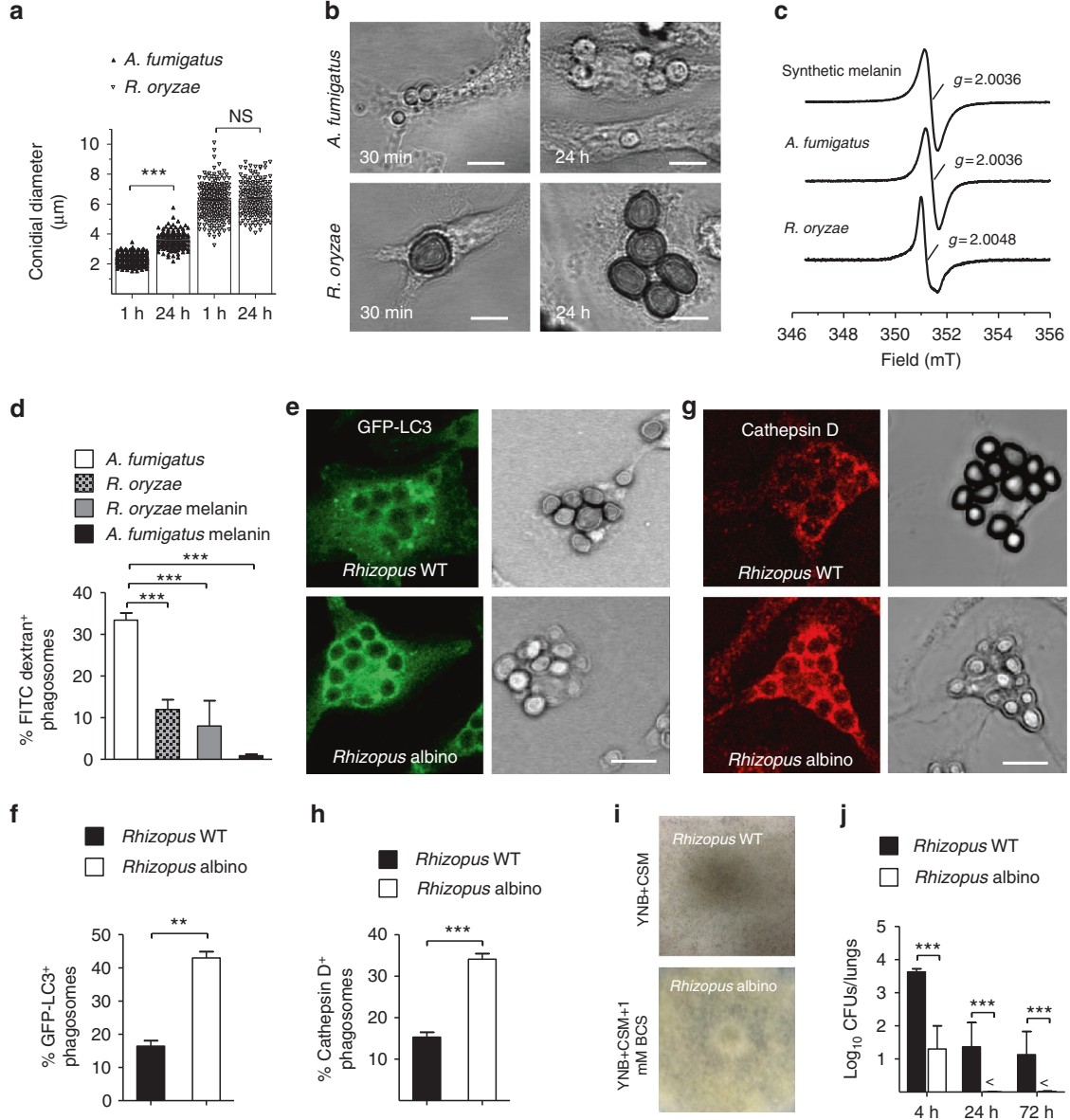

**Fig. 5** *Rhizopus* melanin blocks early event in phagosome biogenesis. **a** GFP-LC3 BMDMs were infected at 1 or 24 h with live conidia of *A. fumigatus* or *R. oryzae* at an MOI 1:1. Cells were fixed and the conidial diameters of intracellular conidia was measured by confocal microscopy. Data on quantification of conidial diameter are presented from one out of three independent experiments. Each symbol represents the value of maximum diameter of individual fungal cell and horizontal bars represent the mean diameter. ***$P < 0.0001$, Mann–Whitney test. **b** Representative DIC images from **a** are shown. **c** X-band room temperature EPR spectra of purified melanin obtained from *A. fumigatus*, *R. oryzae*, or synthetic melanin are shown. **d** BMDMs pre-loaded with FITC-Dextran were stimulated with conidia of *A. fumigatus* or *R. oryzae* or melanin ghosts (purified melanin particles) obtained from conidia of the indicated *A. fumigatus* or *R. oryzae* strains at an MOI of 3:1 (effector:fungal conidia). Cells were removed at 4 h and assessed by confocal imaging. Data on quantification of FITC-Dextran+ phagosomes are presented as mean ± SEM of three independent experiments. **e**, **h** BMDMs from GFP-LC3 (**e**, **f**) or C57BL/ 6 (B6) (**g**, **h**) mice were infected with *R. oryzae* conidia (WT conidia) or *R. oryzae* conidia following chemical degradation of melanin with $H_2O_2$ bleaching (albino conidia) at an MOI of 3:1 (effector:fungal cells). Cells were removed at 1 h of infection, fixed, stained, and analyzed by confocal imaging. Data on quantification of GFP-LC3+ (**f**) or Cathepsin D+ (**h**) phagosomes are presented as mean ± SEM of three independent experiments. **e**, **g** Representative fluorescent images from experiments on **f** and **h** are shown. **i**, **j** Fungal loads from lungs of immunocompetent mice ($n = 9$ per experimental group) infected with $10^6$ conidia of *R. delemar* grown in regular media (WT *Rhizopus*) or under conditions of copper starvation to inhibit melanization (albino *Rhizopus*) (**i**). Mice ($n = 3$ per condition) were sacrificed at the indicated time points, lungs were homogenized, and fungal loads were assessed by CFU plating (**j**). ***$P < 0.0001$,
**$P < 0.001$ Mann–Whitney test. Scale bar, 10 μm

**Iron starvation governs *Rhizopus*–macrophages interplay.** Because of the blockade in P fusion induced by *Rhizopus*, we reasoned that nutritional immunity is the main effector mechanism to inhibit fungal growth inside macrophages. In view of the central role of iron in Mucorales growth[7,8], we performed transcriptomic analysis of host and pathogen

genes to explore whether an iron restriction mechanism accounts for inhibition of conidial growth inside macrophages. Specifically, we performed RNA-seq on poly(A)-enriched RNA isolated from BMDMs infected with *R. delemar* (strain 99-880) at different time points (1, 4, or 18 h). The RNA preparations contained a mixture of messenger RNAs (mRNAs)

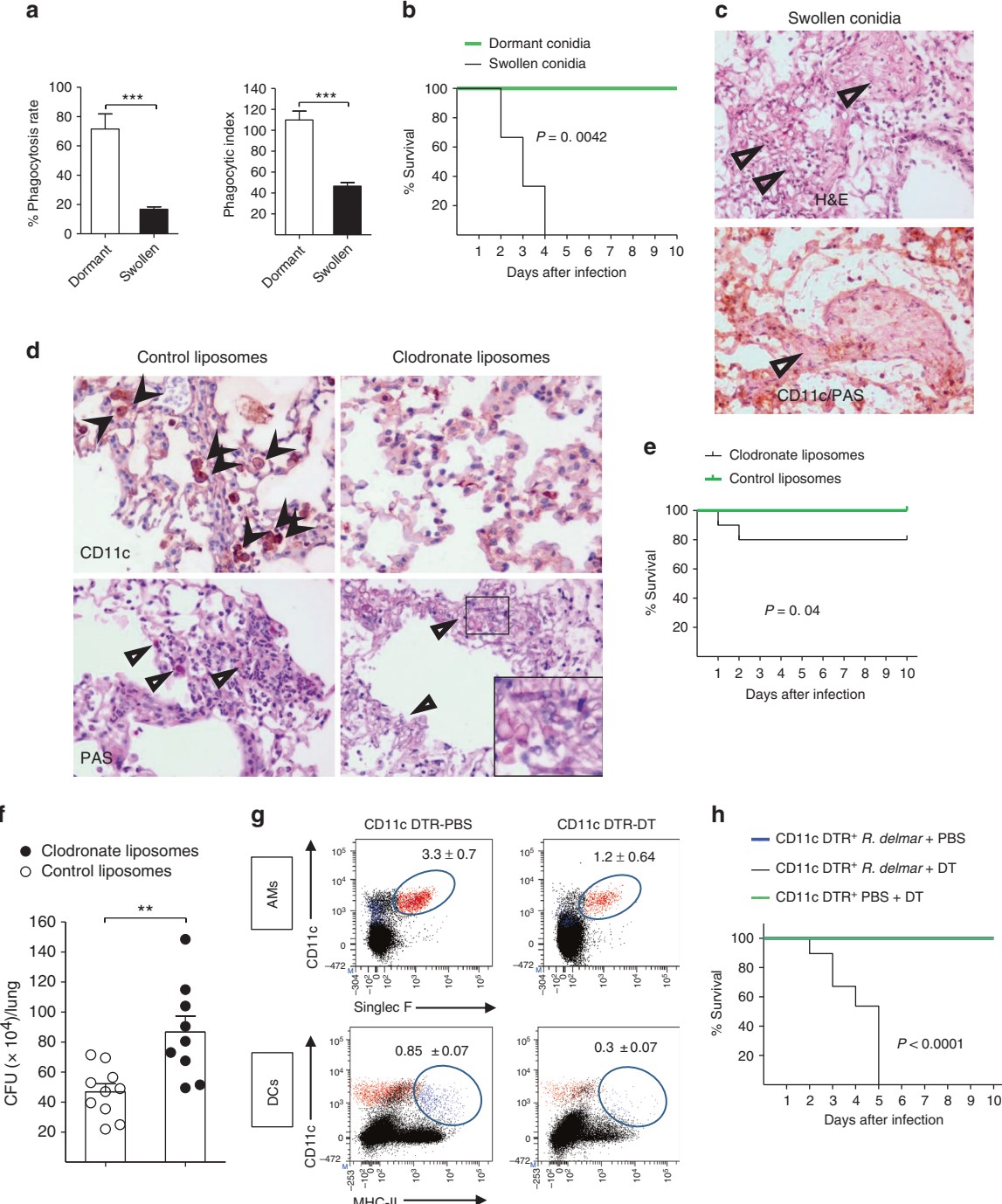

**Fig. 6** AMs have a non-redundant role in immunity against *Rhizopus*. **a** GFP-LC3 BMDMs were infected with dormant or swollen conidia of *R. oryzae* at an MOI of 3:1 (effector:fungal cells) and phagocytosis was assessed at 4 h. Data on quantification of phagocytosis of *Rhizopus* conidia are presented as mean ± SEM of three independent experiments. ***$P < 0.0001$, Mann–Whitney test. **b** Survival of immunocompetent C57BL/6 (B6) mice ($n = 8$ per group) infected via intratracheal administration of a standardized inoculum ($5 \times 10^6$ conidia per mice) of dormant or swollen conidia of *R. oryzae*. **c** Representative photomicrographs of the lungs from mice infected with $5 \times 10^6$ swollen conidia of *R. oryzae* at 24 h of infection. Lungs were stained by H&E or IHC for CD11 and counterstaining with hematoxylin and PAS. **d** Representative histopathology from lungs of C57BL/6 (B6) mice following intratracheal administration of 100 μl of clodronate liposomes or control liposomes and 48 h later infection with $10^7$ *R. oryzae* conidia. Invasive hyphal growth is present in the lungs of clodronate liposome group of mice. **c**, **d** Original magnification ×400. **e** Survival of C57BL/6 (B6) mice treated with clodronate liposomes ($n = 10$) or control liposomes ($n = 15$) and infected as in **d**. **f** Fungal loads in the lungs of C57BL/6 (B6) mice treated with clodronate liposomes or control liposomes and infected as in **d**. Lungs were removed on day 2 of infection and fungal loads were assessed by CFU plating. **$P < 0.001$, Mann–Whitney test. **g** CD11c-DTR mice were intranasally inoculated with 40 μl PBS or with 20 ng/g of body weight of diphtheria toxin in 40 μl PBS and analyzed for the presence of CD11c+ cells (AMs and DCs) at 24 h. Results represent two to three experiments with two to three mice per group per experiment. **h** Survival of CD11c-DTR mice ($n = 5$ per group) treated with DT (20 ng/kg of mice) or PBS (control) and infected 48 h later with $5 \times 10^6$ conidia of *R. delemar*. The nonparametric log-rank test was used to determine differences in survival times

expressed by the *Rhizopus* spp., as well as by the host cells. To ensure that the observed fungal gene expression changes were due to the interaction with the host cell and not simply a response to the medium, we performed RNA-seq on control *Rhizopus* conidia incubated for 5 min in culture medium in the absence of host cells. Similarly, we performed RNA-seq on control, uninfected BMDMs.

From the pathogen perspective, the majority of genes previously implicated in iron acquisition were regulated during infections (Fig. 7a). In particular, we identified *Fet3* and *Ftr1* as the most highly upregulated genes during intracellular persistence of *R. delemar* inside BMDMs. Notably, Fet3, a multicopper ferroxidase required for ferrous iron uptake and its interacting partner, the high affinity iron permease Ftr1p, comprise the major iron assimilation pathway of *Rhizopus* that is induced under iron-limited conditions and has a major pathogenetic role during in vivo infection[9].

In BMDMs, a significant number of iron metabolism-related genes identified in previous transcriptomic studies were differentially expressed over the course of *Rhizopus* infection[28,29], with an expression pattern consistent with activation of an M2-alternative program (Fig. 7b). Specifically, there was evidence of downregulation of typical M1-related genes (*Nos2*, *Mpo*, *Hif1a*, *Egln3*) and upregulation of M2-related genes (*Myc*, *fxn*, *Cyp1b1*, *Alas1*). Additionally, several genes directly linked to trafficking and intracellular distribution of iron inside macrophages (*heph*, *fxn*, *bdh2*, *hfe*, *ltf*, *lcn2*, *Steap 1*, *Steap 2*, *Steap 4*) were differentially expressed during Mucorales infection. On the other hand, *Rhizopus* infection induced downregulation of hemochromatosis (*hfe*) and *Ptgs2* genes, which are both upregulated during alternative activation in macrophages[28–30]. Collectively, dual RNA-seq revealed the induction of an iron restriction response during the course of *Rhizopus* infection in BMDMs.

**Iron restriction inhibits *Rhizopus* growth inside macrophages.** Next, we tested directly whether iron restriction is the primary mechanism of inhibition of fungal growth in macrophages. Importantly, BMDMs were infected with *Rhizopus* with or without the presence of iron, DFO, or iron plus DFO in culture media and germination of intracellular conidia was assessed at different time points of infection. Notably, whereas there was minimal evidence of germination of *Rhizopus* conidia in control-untreated BMDMs, ex vivo supplementation of media with iron, DFO, or both resulted in a significant increase in germination of intracellular conidia that was apparent at 12 h of infection (Fig. 8a, b) and subsequent lysis of macrophages.

As an extra proof of the important role of iron restriction in Mucorales inhibition by macrophages, we tested the ability of *Rhizopus* mutants defective in pathways of iron assimilation, which have been shown to be attenuated in virulence in mouse models of mucormycosis of diabetic ketoacidosis and neutropenia, to germinate intracellularly following iron supplementation[9,10]. In particular, we evaluated interaction of macrophages with *ftr1* attenuated mutant of *Rhizopus* spp., which is compromised in virulence because of its diminished capacity to grow under iron-limiting conditions and was highly induced in transcriptomics analysis[9]. Although *Rhizopus* mutant with reduced *FTR1* copies was not killed by macrophages, it displayed major defects in germination following iron supplementation by either FeCl₃, DFO, or both (Fig. 8c, d). Notably, infection of BMDMs with the *Rhizopus fob1/2* mutant with defect in DFO uptake[10] displayed impaired germination following supplementation of culture media with DFO, which was by-passed in the presence of exogenous iron (Fig. 8e, f). Collectively, these findings clearly reveal the essential role of iron restriction inside the phagosome of macrophages on inhibition of intracellular growth of Mucorales conidia.

**Persister conidia inside macrophages in human mucormycosis.** In order to explore the physiological relevance of our studies in humans, we performed histopathological analysis of surgical specimens obtained from a patient with acute myelogenous leukemia who developed disseminated mucormycosis. The patient underwent radical surgery including splenectomy because of worsening necrotizing pneumonia with infiltration of the chest wall and the spleen (Fig. 9a). In histopathology, there was evidence of necrotizing angioinvasive growth of Mucorales hyphae along with areas of granulomatous inflammation in the spleen that contained intracellular Mucorales conidia inside macrophages (Fig. 9b–e). Collectively, these findings are consistent with a pathogenetic model of mucormycosis highlighting the central role of Mucorales–macrophage interplay in disease development (Fig. 10).

**Discussion**

Mucorales are successful pathogens of a range of phylogenetically disparate hosts, from plants to invertebrates and humans[7,8,31–33]. The emergence of mucormycosis in patients with acquired innate immune defects illustrates the important role of professional phagocytic cells in Mucorales host defense[8]. In contrast, the rarity of mucormycosis in patients with primary immunodeficiencies, who are prone to infections caused by other filamentous fungi[34–36], implies that distinct immune pathways are important to restrict Mucorales growth. For example, *Aspergillus* is the primary pulmonary pathogen in patients with genetic defects in NADPH oxidase (chronic granulomatous disease; CGD)[1,16]. In contrast, mucormycosis is a rare infection in CGD patients and occurs only upon administration of corticosteroids[36].

Herein, we reveal the essential role of *Rhizopus*–macrophage interplay for infection outcome and introduce evidence that a central pathogenetic event in development of mucormycosis is related to the prolonged intracellular survival of the fungus inside these immune cells. In addition, we dissect the molecular mechanisms that allow *Rhizopus* to persist inside macrophages via melanin-induced phagosome maturation arrest. Finally, we identify nutritional immunity via iron restriction inside the phagosome as an important host defense mechanism during pulmonary mucormycosis. These findings lead to a pathogenetic model of mucormycosis that links abnormalities in iron metabolism with nutritional immunity inside macrophages and has important implications in future design of therapeutics against mucormycosis.

Our findings on prolonged persistence of *Rhizopus* in the lungs of immunocompetent mice are in sharp contrast to the rapid clearance of *Aspergillus* (this study and refs.[22,26]) and in line with the increased pathogenicity of Mucorales. Previous studies describe the fungistatic activity of macrophages and other professional phagocytes against *Rhizopus* as a possible link to their increased virulence[32,33,37–40]. The present work extends on these findings by providing mechanistic insight on the molecular aspects of macrophages–Mucorales interplay. Specifically, our in vivo studies clearly demonstrate that *Rhizopus* conidia display a tropism and establish prolonged intracellular survival inside AMs that accounts for persistence in the lungs of immunocompetent mice. Importantly, definitive evidence on the non-redundant role of macrophages and other CD11c+ cells of the lungs in immunity against *Rhizopus* was shown by in vivo studies following targeted depletion with the use of clodronate liposomes and CD11c-DTR transgenic mice. In addition, persister conidia of Mucorales inside macrophages were found in a patient with disseminated mucormycosis. Notably, evidence of Mucorales conidia in tissue has also been reported as a characteristic histopathological finding of mucormycosis[41]. These findings have

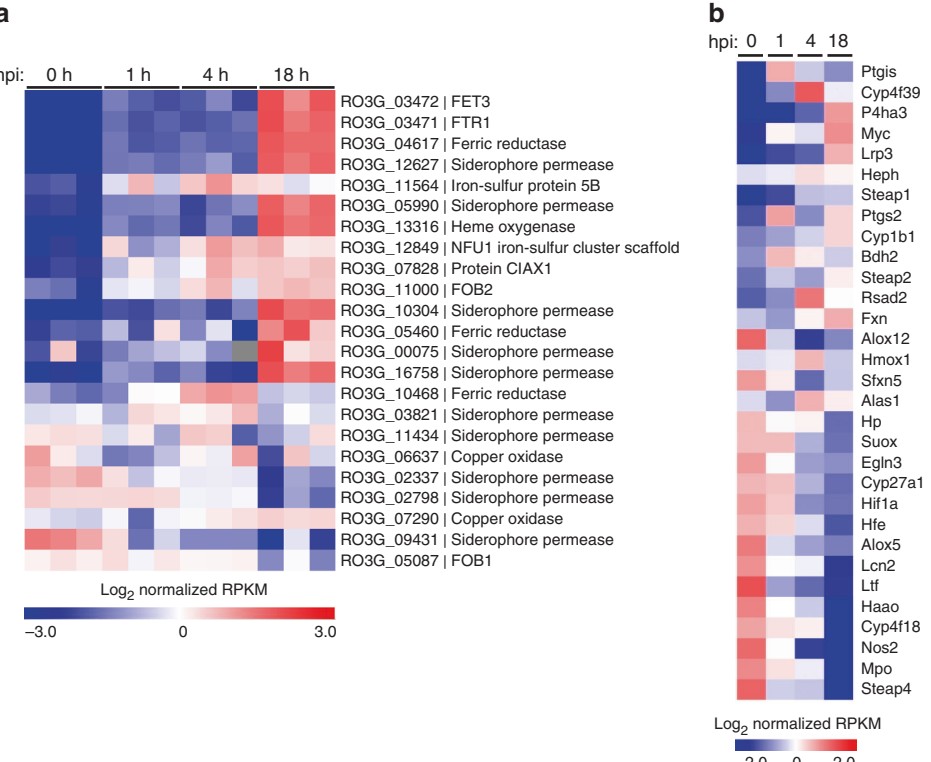

**Fig. 7** Global analysis of differential gene expression during infection. RNA-seq-based expression analysis of iron-related **a** *R. delemar* genes and **b** mouse genes following in vitro infection of BMDMs. **a** Each column represents an individual sample (biological triplicates of four different conditions; $n = 12$). Log-transformed absolute expression normalized across all samples. Red indicates high gene expression. Blue indicates low gene expression. For the 0 h column, *R. delemar* was incubated in tissue culture media without BMDMs for 1 min. **b** Values are presented as in **a** and represent the average of three biological replicates for each condition. For the 0 h columns, BMDMs were incubated in culture media in the absence of *R. delemar* spores for 1 hr

important implications in pathogenesis, epidemiology, and therapeutics of mucormycosis. Specifically, the ability of a classic extracellular pathogen to remain in intracellular dormancy could provide protection from host effector mechanisms and antimicrobial therapy[42]. Notably, failure to eradicate intracellular conidia of Mucorales might also explain clinical observations on relapse of mucormycosis years following cessation of secondary antifungal prophylaxis[43].

In addition, our studies on biogenesis of *Rhizopus* phagosome identified the mechanism of resistance of conidia to killing by macrophages. Specifically, we demonstrate that *Rhizopus* conidia are fully susceptible to oxidative and non-oxidative effector mechanisms of macrophages. In contrast to recent work on macrophage cell lines showing germination and lysis of host cells by conidia of *Mucor circinelloides*[20], we show that primary macrophages completely inhibit Mucorales growth. These results are consistent with studies showing that bronchoalveolar macrophages harvested from lungs of immunocompetent mice are able to ingest and inhibit germination of *R. oryzae* spores without killing them in vitro[40]. The inability of macrophages to kill *Rhizopus* is due to inhibition of LAP, a specialized pathway of phagosome biogenesis with central role in regulation of immune homeostasis and antifungal host defense[21,22]. These early inhibition effects result in phagosome maturation arrest and account for resistance to macrophage killing.

A fundamental difference in biogenesis of *Rhizopus* vs. *Aspergillus* phagosomes that emerged from our study is the lack of intracellular swelling of *Rhizopus* conidia, which accounts for prolonged surface retention of cell wall melanin and subsequent inhibition of LAP and phagosome responses[22]. Furthermore, the ability of melanin to inhibit apoptosis in macrophages via

sustained activation of Akt/PI3K signaling[44] facilitates the establishment of prolonged intracellular dormancy. In view of the critical role of melanin in intracellular survival of *Rhizopus*, further understanding of melanin biosynthesis pathway in these pathogens and harnessing activation of LAP pathway could pave the way for alternative therapeutic strategies in mucormycosis that will prevent or treat the disease with reduced risk of relapse.

In vivo studies on infection of immunocompetent mice with dormant vs. swollen conidia provided insight on the critical role of nutritional immunity mechanisms during early stages of infection by *Rhizopus*. Specifically, the inability of host related factors to inhibit *Rhizopus* growth results in failure of physiological immune responses to prevent invasive fungal growth and acute lethality, as evidenced by infection of immunocompetent mice with swollen conidia. Therefore, poorly understood intracellular and extracellular nutritional immunity mechanisms are at the first line of defense against Mucorales. One clear critical aspect of the failure of nutritional immunity responses against mucormycosis is clinically manifested by the unique susceptibility of diabetic ketoacidosis and DFO-treated patients (DFO is utilized by Mucorales as a xenosiderophore to supply iron to the fungus) to mucormycosis[6–8]. These two patient categories suffer from elevated available serum iron[11,45]. Our transcriptomic analysis of host and fungal iron regulated genes during the course of infection is consistent with activation of an iron restriction response in macrophages triggering the induction of iron starvation pathways in Mucorales, with notable example being the upregulation of the high affinity iron permease *FTR1*, its interacting partner *FET3*, and genes encoding ferric reductases[9]. Furthermore, direct evidence on the central role of iron restriction in inhibition of Mucorales growth is provided by reversal of

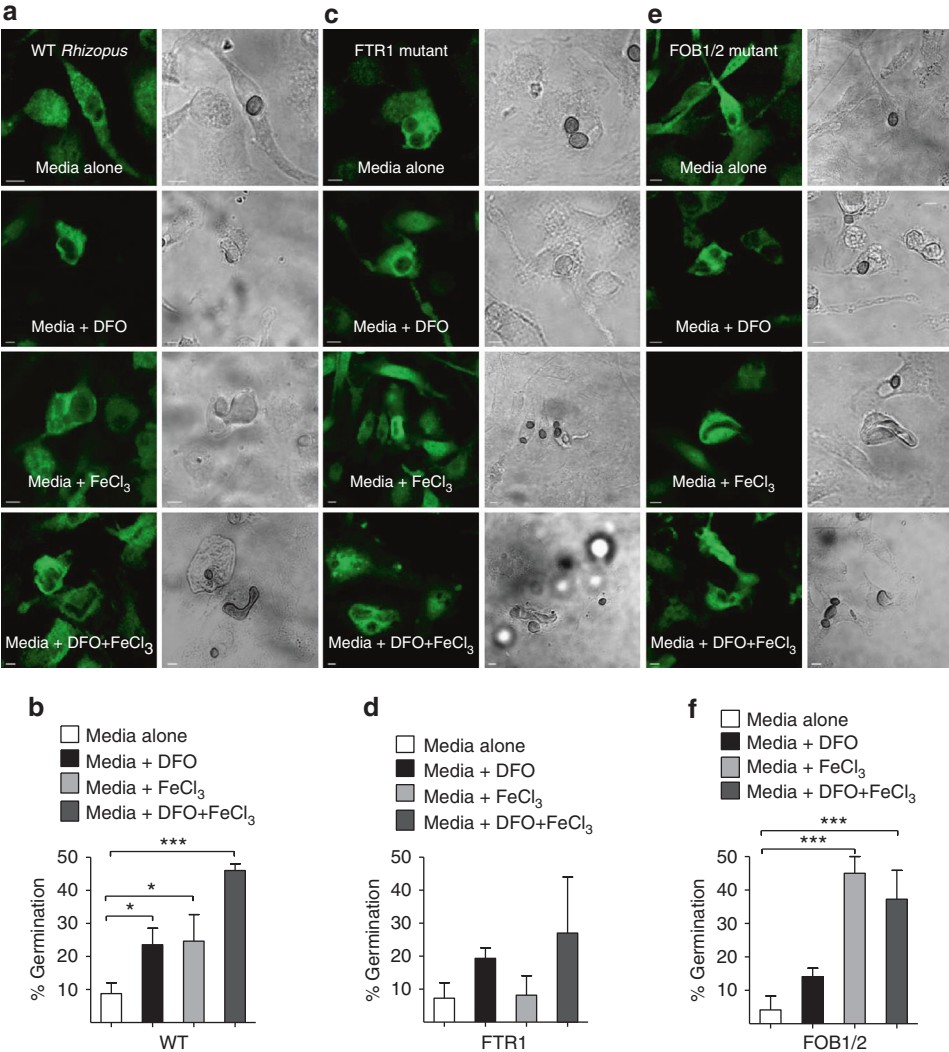

**Fig. 8** Macrophages inhibit intracellular growth of *Rhizopus* via iron starvation. **a**, **b** BMDMs from GFP-LC3 mice were infected with WT *Rhizopus delemar* strain at an MOI of 1:2 (effector:fungal cells) in regular culture media or media containing DFO (100 μM), iron (FeCl$_3$; 100 μM), or DFO plus iron. At 1 h of infection, cells were extensively washed to remove non-phagocytosed conidia, media were replaced, and intracellular germination was assessed at 12 h by confocal imaging. BMDMs from GFP-LC3 mice were also infected with *R. delemar*-attenuated mutants for *FTR1* (**c**, **d**) or FOB1/2 (**e**, **f**), which are defective in the high affinity iron permease expression regulating iron assimilation under conditions of limited iron availability and DFO receptor FOB1/2 mediating fungal iron uptake from DFO, respectively, as in **a**. The attenuated mutants were derived from the wild-type strain used in **a**, **b** by RNAi[9, 10]. Representative immunofluorescence images are shown in **a**, **c**, and **f**. Data on quantification of germination of intracellular conidia of the indicated *Rhizopus* strain are presented as mean ± SEM of three independent experiments. ***$P < 0.0001$, *$P < 0.01$, Mann–Whitney test. Bar, 5 μm

the fungal growth inhibition when iron was exogenously administered in the form of FeCl$_3$, DFO, or ferrioxamine (the iron-rich form of DFO). Finally, studies utilizing *Rhizopus ftr1* mutant, which displays genetic defects in iron assimilation pathways and attenuated pathogenicity[9,10], showed inability to germinate inside the macrophages in the presence of iron supplementation. Our studies on the role of iron homeostasis inside macrophages in antifungal host defense should foster the concept of targeted iron chelation therapies inside these cells[46].

Overall, our study provides a pathogenetic model of mucormycosis, which places nutritional immunity inside macrophages in the forefront of antifungal immunity in the lung and explains how abnormalities in iron metabolism lead to development of immunodeficiency. Furthermore, these findings pave the way for future studies on host determinants of iron homeostasis in macrophages implicated in development of invasive fungal pneumonia[47,48]. Several questions on the signaling pathways, cytokine responses, and molecular components of iron

homeostasis inside macrophages that maintain iron restriction inside the fungal phagosome remain unanswered. Notably, in patients with immune defects in LAP and/or other signaling pathways regulating phagolysosomal fusion, restriction of iron availability inside phagocytes could become the predominant effector mechanism to prevent invasive fungal disease. Therefore, dissecting abnormalities of iron metabolism in myeloid cells of immunocompromised patients at high risk for invasive mold infections should become a priority for development of future therapeutic strategies. Finally, the role of intracellular and extracellular nutritional immunity mechanisms related to other metals or nutrients against medically important fungi deserve further exploitation.

## Methods
**Reagents**. The following antibodies and reagents were used for ex vivo studies in murine primary macrophages: anti-GFP polyclonal Ab (Minotech, #721-1, dilution 1:500), Cathepsin D (Santa Cruz, #sc-377299, dilution 1:100), Rab5B (clone A20,

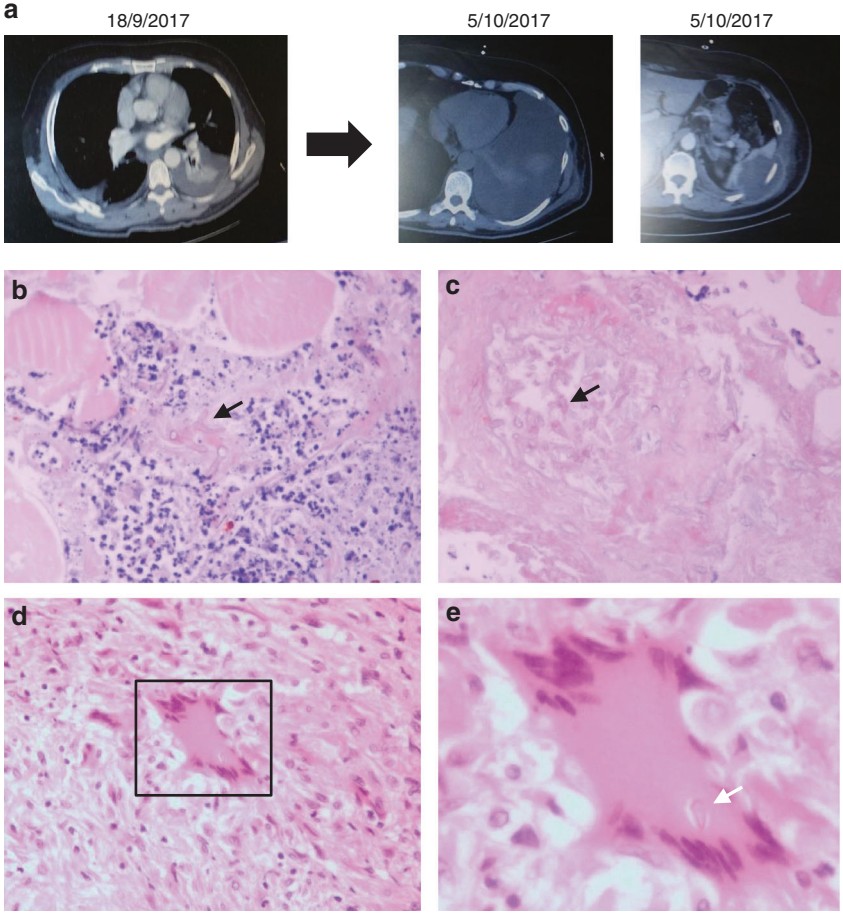

**Fig. 9** Invasive pulmonary mucormycosis in a patient with intracellular persistence (**a**). A 60-year-old white male with relapsed acute myelogenous leukemia received salvage chemotherapy with fludarabine and high-hose cytarabine (FLAG) in August 2017. On September 2017, the patient developed febrile neutropenia, left-sided chest pain, and evidence of necrotizing pneumonia with chest wall myositis in computer tomography of the chest. He received treatment with broad-spectrum antibiotics and anidulafungin. Despite neutrophil recovery, the infection progressed with infiltration of the abdominal wall and the spleen. In tissue biopsy mucormycosis was diagnosed based on characteristic histopathological findings (**b**, **c**). The patient was started on liposomal amphotericin B and radical surgery with splenectomy was performed in November 2017. In histopathological analysis of muscle biopsy, there is evidence of extensive tissue necrosis and growth of Mucorales hyphae inside blood vessels (**b**, **c**). In spleen histopathology there are areas of granulomatous lesions containing macrophages and multinucleated giant cells (**d**). **e** In higher magnification (**d**, inset), there is evidence of a Mucorales conidium (spore) inside a giant cell (arrow). Original magnifications ×200, ×800. Eosin and hematoxylin stain; white arrows: spores; black arrows: hyphae

Santa Cruz, # sc-598, dilution 1:100), Alexa 555 (Life Technologies, #A21425, dilution 1:1000), Alexa 488 (CF488A, Biotium, #20012-1, dilution 1:1000), FITC Annexin V Apoptosis Detection Kit (BD Pharmingen, #556547), FITC-Dextran (Sigma, #46945), propidium iodide (Sigma, #P4170), DFO (Sigma, #D9553), FeCl$_3$ (Sigma, #157740), clodronate liposomes (http://www.clodronateliposomes.org), bathocuproinedisulfonic acid disodium salt (BCS) (Sigma-Aldrich, #B1125), and RNeasy Plant Mini Kit (Qiagen, #74903).

**Microorganisms and culture conditions.** *Aspergillus fumigatus* ATCC[22] and *Rhizopus* strains used (WT *R. oryzae* ATCC557969[32]; WT strain *R. delemar* 99-880, a brain isolate obtained from the University of Texas Health Science Center at San Antonio, which had its genome sequenced[9,10]) were grown on Yeast extract agar glucose agar plates for 3 days at 37 °C. *Rhizopus delemar* M16 is a *pyrf*-null mutant that is derived from *R. delemar* 99-880 and is unable to synthesize its own uracil[49], and was grown on YPD medium (MP Biomedicals) supplemented with 100 µg/ml uracil. *Rhizopus delemar* with reduced *FTR1* copy number, and *R. delemar* transformed with RNA interference (RNAi) targeting *FTR1* expression were all derived from strain M16[9]. In experiments including RNAi mutants, a chemically defined synthetic medium containing yeast nitrogen base (YNB) supplemented with complete supplemental mixture without uracil (CSM-URA) (MP Biomedicals) (i.e., YNB + CSM-URA) (formulation per liter, 17 g YNB without amino acids (YNB) (BD), 20 g dextrose, and 7.7 g complete supplemental mixture minus uracil) was used. Fungal conidia (spores) were harvested by gentle shaking in the presence of sterile 0.1% Tween-20 in phosphate-buffered saline (PBS), washed twice with PBS, filtered through a 40 µm pore size cell strainer (Falcon) to separate conidia from contaminating mycelium, counted by a hemocytometer, and suspended at a concentration of 10$^7$ and 10$^8$ spores/ml for *Rhizopus* and *Aspergillus* strains, respectively. Inactivation of *Rhizopus* conidia was done by exposure to UV light (1 h, room temperature). To achieve synchronized swelling of *Rhizopus* conidia, 10$^6$/ml dormant conidia were incubated at 28 °C in a 6-well plate with RPMI-MOPS supplemented with 2% glucose for 4 h. For fluorescence labeling of *Rhizopus*, 10$^6$ conidia were stained in 100 µl PBS containing 100 µg/ml Fluorescent Brightener 28 (Sigma-Aldrich, #475300) and 0.1 M NaHCO$_3$ for 30 min protected from light in a bench-top rotator[33]. Afterwards, conidia were washed three times with PBS and the concentration was adjusted to 10$^7$ or 10$^8$ conidia/ml.

**Fungal melanin extraction.** The isolation of melanin from *A. fumigatus* and *R. oryzae* conidia was performed[22]. Briefly, conidia were treated with a combination of proteolytic (proteinase K; Sigma) and glycohydrolytic (Glucanex; Novo) enzymes, denaturing guanidine thiocyanate, and hot concentrated HCl (6 M). This treatment resulted in an electron-dense layer similar in size and shape to the original conidial melanin layer without underlying cell components, for which reason these electron-dense materials were called melanin ghosts.

**Chemical characterization of *R. oryzae* melanin.** *Rhizopus* conidia of 1.6 g was ground with mortar and pestle under liquid nitrogen into very fine powder. The black fine powder was extracted by boiling with 5% KOH under reflux for 1 h followed by filtration. The black colored filtrate was left to cool at room temperature and then precipitated with 1 N HCl. The black precipitate was collected by filtration using filter paper. The black precipitate was left to dry on the filter paper

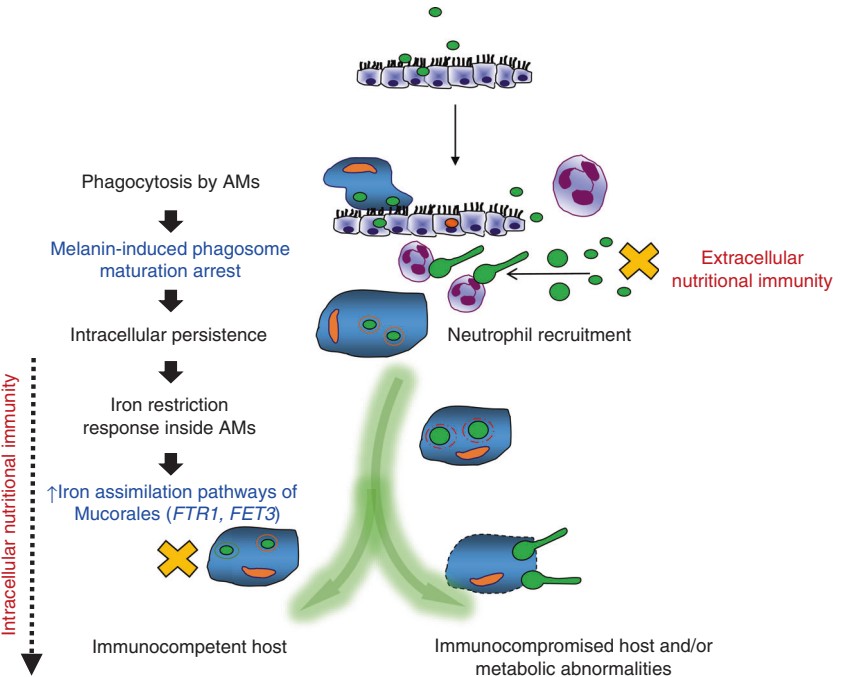

**Fig. 10** Proposed model of nutritional immunity inside macrophages against Mucorales. Following inhalation, Mucorales conidia are predominantly phagocytosed by alveolar macrophages (AMs). Intracellular conidia of Mucorales remain dormant and establish prolonged persistence inside the nascent phagosome of AMs. Surface retention of cell wall melanin in intracellular conidia of Mucorales blocks phagosome biogenesis and LAP, impedes killing, and induces anti-apoptotic signaling in macrophages to establish persistence. Inhibition of intracellular germination of Mucorales conidia via iron restriction is a central host defense mechanism against mucormycosis. In addition, incompletely understood nutritional immunity mechanisms, including transferrin-mediated restriction of free iron availability in serum, inhibit germination of extracellular conidia. In parallel, rapid recruitment of neutrophils results in clearance of extracellular conidia of Mucorales. Quantitative and qualitative defects in innate immunity associated with prolonged chemotherapy-induced neutropenia, and/or corticosteroid-induced immunosuppression compromise the ability of phagocytes to inhibit germination of intracellular or extracellular conidia and result in invasive fungal growth. On the other site, failure of nutritional immunity mechanisms in patients with abnormalities on iron metabolism (e.g., diabetic acidosis) allow germination of intracellular or extracellular conidia and promote invasive tissue growth. Mucorales responses inside AMs are highlighted in blue. Nutritional immunity responses are highlighted in red

followed by washing several times with 1 N HCl, then 3 N HCl, water, and methanol. The remaining black/brown precipitate was dried at room temperature and used for further analysis.

The chemical analysis of melanin pigment was carried out by the modified method of Fava et al.[50]. Briefly, the solubility of the black pigment in distilled deionized water, 0.1 N HCl, 1 N HCl, 3 N HCl, 1 N KOH, methanol, ethanol, acetone, chloroform, benzene, and dimethyl sulfoxide (DMSO) were checked and found to be insoluble in all these solvents with the exception of 1 N KOH (Supplementary Table 1). Reactions with oxidizing agents such as 6% sodium hypochlorite and 30% $H_2O_2$ were determined by measuring solubility of the pigments in these reagents[50]. The precipitation of the pigments with $FeCl_3$ which reacts to polyphenols was also tested and found to be precipitated (Supplementary Table 1).

**UV absorbance and IR analysis of *Rhizopus* melanin**. The melanin extract in 1 N KOH was measured at 200–700 nm with the use of a SPECTRO UV–VIS spectrophotometer. One normal KOH was used as blank. For IR measurement, melanin powder was mixed with KBr and used to measure IR absorbance using a Bruker machine with KBr disc used as a blank.

**Melanin alkaline $H_2O_2$ oxidation**. To identify the production of various pyrrole acids (PTCA, PDCA, isoPTCA, and PTeCA) from melanin samples by LC-MS, alkaline $H_2O_2$ degradation was performed as described by Ito et al.[51]. In brief, extracted *Rhizopus* melanin was taken in a 5-ml screw-capped conical test tube, to which 500 μl of 1 mol/L $K_2CO_3$ and 50 μl of 30% $H_2O_2$ (final concentration: 1.5%) were added. The mixture was vigorously mixed and then kept on a shaker at 25 °C for 20 h. The residual $H_2O_2$ was decomposed by adding 100 μl 10% $Na_2SO_3$ and the mixture was then acidified with the addition of 200 μl of 6 M HCl. The reaction mixture was centrifuged at $4000 \times g$ for 5 min and then subjected to thin linear chromatography (TLC) along with total KOH extract. The TLC developing system comprised of water:1 M HCl:chloroform:methanol (0.5:0.5:1:6).

**LC-MS of melanin hydrolyses product**. LC-MS analyses were carried out in negative ion mode by electrospray ionization on (Waters) ACQUITY UPLC triple Quadruple (Xevo TQD) instrument equipped with the MassLynx software, at a flow rate of 0.3 ml/min, run time of 5 min, and the use of a solvent system containing 85% methanol, 15% water, and 0.1% formic acid. All solvents and reagents were HPLC grade and used without further purification.

**EPR studies**. EPR spectra were recorded at room temperature on an ELEXSYS Bruker spectrometer equipped with a Super High Q Sensitivity resonator operating at X-band (9.9 GHz). Microwave power was 1 mW for synthetic melanin and 0.1 mW for *A. fumigatus* or *R. oryzae*. Magnetic field modulation amplitude and frequency were, respectively, set to 0.2 mT and 100 kHz.

**Virulence studies in mice**. GFP-LC3 (obtained from RIKEN BioResource Center) and C57BL/6 (B6) mice were maintained in grouped cages in a high-efficiency particulate air-filtered environmentally controlled virus-free facility (24 °C, 12/12-h light/dark cycle), and fed by standard chow diet and water ad libitum. All experiments were approved by the local ethics committee of the University of Crete Medical School, Greece in line with the corresponding National and European Union legislation. Animal studies on virulence of albino *Rhizopus* were approved by the IACUC of the Los Angeles Biomedical Research Institute at Harbor-UCLA Medical Center, according to the NIH guidelines for animal housing and care.

For virulence studies, 8- to 12-week-old female C57BL/6 (B6) mice were challenged by intratracheal installation with a standard dose of *A. fumigatus* or *Rhizopus* conidia. Mice were euthanized at the indicated time point, lungs were homogenized, and CFU counts were assessed[21,22]. For AM depletion studies, mice received by intratracheal administration 100 μl of clodronate liposomes (containing 500 μg of clodronate; http://www.clodronateliposomes.org) or control (empty) liposomes[26]. For CD11c cell depletion, CD11c-DTR mice received by intratracheal administration 20 ng/kg of diphtheria toxin (DT). The efficiency of cell depletion was assessed by immunohistochemistry for CD11c and flow cytometry analysis of bronchoalveolar lavage (see below).

Immunocompetent CD-1 male mice were infected with $10^6$ spores of *R. delemar* strain 99-880 grown in two different condition: on synthetic defined (SD) medium complete plates and on SD complete plates + 1 mM of the copper chelator, BCS. Sporulation on plates with BCS was subjected to copper deprivation and thus conidia appeared pigmentless (referred to in experiments as albino). Infection was carried out intratracheally, with 9 mice per group. Lungs were collected at three different time points: 4 h post infection, at day +1, and at day +3. At each time point, three mice per group were sacrificed. Right after infection, two mice have been sacrificed for inoculum verification. After collection, the samples were plated on potato dextrose agar + 0.1% Triton plates and incubated at 37 °C. For lung samples, after homogenization in 2 ml of PBS, 200 µl were plated directly from the concentrated samples and also from serial dilutions to facilitate counting.

**Generation of murine BMDMs.** BMDMs were generated by culturing BM cells obtained from 8- to 12-week-old female mice in Dulbecco's modified Eagle's medium (DMEM), supplemented with L929 cell-conditioned medium (30%). The resulting cultures consisted of macrophages (>95% purity), as determined by staining for F4/80 and flow cytometry.

**Isolation and stimulation of BMDMs.** The BMDMs were collected and resuspended in DMEM culture medium supplemented with streptomycin 1% and fetal bovine serum (FBS) 10% (DMEM complete). The cells were counted in a Bürker counting chamber, and their number was adjusted to $10^6$/ml. For immunofluorescence experiments, a total of $1 \times 10^5$ BMDMs per condition in a final volume of 100 µl were allowed to adhere to polylysine-treated glass coverslips (Ø 12 mm) for 1 h followed by stimulation with conidia of *Rhizopus* or *A. fumigatus* at a multiplicity of infection (MOI) of 2:1 and 5:1, respectively, at 37 °C in a 5% $CO_2$ incubator for the indicated time point. After infection, cells were washed twice with PBS to remove medium and non-phagocytosed spores and cells were fixed on the coverslips for 15 min in 4% paraformaldehyde, followed by 10 min fixation with 100% ice-cold methanol and then stored in PBS at 4°C until immunofluorescence staining.

**Immunofluorescence staining.** For immunofluorescence imaging, cells were seeded on coverslips pretreated with polylysine, fixed with 4% paraformaldehyde for 15 min at room temperature, and followed by 10 min of fixation with ice-cold methanol at −20 °C. Next, the coverslips were washed twice with PBS, permeabilized by using 0.1% saponin (Sigma-Aldrich) prior to blocking for 30 min in PBS-BSA (PBS + 2% BSA). The coverslips were then incubated for 1 h with the indicated primary antibody (Ab), washed twice in PBS-BSA, then counterstained with the appropriate secondary Alexa Fluor secondary Ab (Molecular Probes), and followed by DNA staining with 10 µM TOPRO-3 iodide (642/661; Invitrogen). After extensive washing, slides were mounted in Prolong Gold antifade media (Molecular Probes). Images were acquired using a laser-scanning spectral confocal microscope (TCS SP2; Leica), LCS Lite software (Leica), and a ×40 Apochromat 1.25 NA oil objective using identical gain settings. A low fluorescence immersion oil (11513859; Leica) was used, and imaging was performed at room temperature. Serial confocal sections at 0.5 µm steps within a z-stack spanning a total thickness of 10 to 12 µm of the cell were taken and 3D images were generated using the LCS Lite software to assess for internalized conidia contained within phagosomes. Unless otherwise stated, mean projections of image stacks were obtained using the LCS Lite software and processed with Adobe Photoshop CS2. Phagosomes surrounded by a rim of fluorescence of the indicated protein marker were scored as positive.

**Lysosomal extract preparation.** Crude BMDM lysosomal extracts were obtained using the lysosome isolation kit instructions (Thermo Scientific, Boston, MA, USA), with small modification of a previously described protocol[52]. Briefly, at least $3 \times 10^8$ freshly collected BMDMs were counted, centrifuged at $400 \times g$ for 5 min at 4 °C and then washed twice with cold PBS. The supernatant was removed, the pellet was resuspended with reagent A and 1% (v/v) protease inhibitor, and then incubated in ice for exactly 2 min. Next, the cells were mildly sonicated on ice for 10 s and checked under a microscope with Tryptan blue solution staining to verify lysis of BMDMs. Subsequently, reagent B with 1% (v/v) protease inhibitor was added and the tube was inverted several times to mix the solution, which was then centrifuged at $500 \times g$ for 10 min at 4 °C. The supernatant was collected and gradient dilution buffer was added. The solution was centrifuged at $18000 \times g$ for 30 min at 4 °C and the pellet was dissolved in 250 µl gradient dilution buffer. Two rounds of sonication for 10 s each were performed, resulting in the generation of crude lysosome extract.

**In vitro studies with crude lysosomal extracts.** Crude lysosomal extracts were added to a 96-well plate in increasing concentration of 10%, 25%, and 50% with culture medium (DMEM-Glutamax supplemented with 10% FBS and 1% streptomycin) at pH 5.5. *Rhizopus oryzae* and *A. fumigatus* conidia were counted and added at a number of $5 \times 10^4$/well, reaching a total volume of 100 µl in each well. Culture medium at pH 5.5 was added to control wells. Plates were incubated for 24 h at 37 °C in a 5% $CO_2$ incubator. The fungal metabolic activity was assessed with XTT ((2,3)-bis (2-methoxy 4-nitro 5-sulfenyl)-2*H*-tetrazolium carboxanilide;

Sigma-Aldrich) reduction assay. One hundred microliters of tetrazolium salt XTT and menadione was added to each well at a final concentration of 0.25 mg/ml and 25 µM, respectively, and the plate was incubated for an additional 1 h. The absorbance of formazan, the XTT reduction product, was read at 450 and 655 nm on a Bio-Rad 680 microplate spectrophotometer. The percentage of the metabolic activity was determined as follows: % metabolic activity = 100% × ($OD_{450}$ − $OD_{655}$) experiment/($OD_{450}$ − $OD_{655}$) control. Fungal killing was evaluated by plating on a Sabouraud agar plate a 100-fold dilution of each well in sterile PBS.

**Electron microscopy studies.** Electron microscopy was performed using the methods reported previously[53]. Briefly, BMDMs were collected, counted, and inoculated in DMEM-Glutamax, 10% FBS, 1% streptomycin in 6-well plates. BMDMs were infected with conidia of either *R. oryzae* or *A. fumigatus* and cells were removed at the indicated time point of infection. Accordingly, infected BMDMs were fixed for 30 min at 4 °C with 2% glutaraldehyde in cold sodium cacodylate buffer (SCB) (0.1 M sodium cacodylate, 0.25 M sucrose, pH 7.4), and washed again with SCB. This was followed by two 30-min incubations in acid phosphatase reaction buffer (0.1 M sodium acetate, 1 mM glycerophosphate, and 2 mM $CeCl_3$), pH 5.2, at 37 °C. The cells were then rinsed three times with acid phosphatase reaction buffer, and re-fixed in 3% glutaraldehyde in SCB for 1 h at 4 °C. After two more washes in SCB, the obtained monolayers were post-fixed in cacodylate-buffered 1% $OsO_4$ for 2 h, dehydrated, and embedded in Epon 812 (Merck, Darmstadt, Germany). An ultratome (Leica, Reichert Ultracuts, Wien, Austria) was used to cut ultrathin sections, which were contrasted with 4% uranyl acetate for 45 min and lead citrate for 4 min at room temperature. Finally, the sections were examined using a Jeol 1200 EX2 electron microscope (JEOL, Tokyo, Japan).

**Phagocytosis and killing assays in BMDMs and PMNs.** For the killing assays, $10^6$ BMDMs were left to adhere in 6-well plates for 1 h in 2 ml of DMEM complete media at 37 °C in a 5% $CO_2$ incubator, and subsequently infected with either *R. oryzae* or *A. fumigatus* conidia, at an MOI of 1:1. BMDMs were washed three times with warm PBS 30 min after the infection to remove non-phagocytosed conidia. At the indicated time point of infection (2 or 6 h), BMDMs were harvested by scraping, placed in Eppendorf tubes, lysed by sonication (three times for 10 s and once for 5 s for *A. fumigatus*-infected and *R. oryzae*-infected BMDMs, respectively), centrifuged at 1000 rpm for 10 min at 4 °C, and the pellet containing intracellular conidia was resuspended in 200 µl sterile PBS. *Aspergillus fumigatus* killing was assessed as previously described using propidium iodide staining[22]. For the evaluation of killing of *Rhizopus* conidia by BMDMs, intracellular conidia recovered after BMDM lysis were incubated at 37 °C in a 5% $CO_2$ incubator with DMEM complete medium for ~4 h, until germination was microscopically observed. Killing of *R. oryzae* was assessed using a Bürker counting chamber based on the percentage of germinating conidia. Germination of intracellular *Rhizopus* conidia by BMDMs was always normalized to the germination of control *R. oryzae* conidia following sonication (5 s) and cultured in DMEM complete medium for ~4 h in the absence of BMDMs, which was typically always >95%. In representative experiments, killing of *A. fumigatus* was also assessed based on the germination assay.

For phagocytosis assay, BMDMs and polymorphonuclear neutrophils (PMNs) from GFP-LC3 mice were stimulated with *R. oryzae* and *A. fumigatus* conidia at an MOI of 2:1 at 37 °C in a 5% $CO_2$ incubator for different time points. Cells were then fixed and stained for confocal microscopy as previously mentioned. Phagocytic index was expressed with the following formula: (total number of engulfed cells/total number of counted macrophages) × (number of macrophages containing engulfed cells/total number of counted macrophages) × 100.

**Murine PMN isolation.** Murine PMNs were isolated using a Percoll (Sigma) double gradient density centrifugation technique. Bone marrow from two immunocompetent GFP-LC3 mice was collected and flushed in room temperature in a sterile solution of PBS/EDTA. The cells were centrifuged at room temperature for 10 min at $350 \times g$ and resuspended in 2 ml PBS/EDTA. The cells were carefully placed on top of 2 ml of three different Percoll concentrations (75%, 67%, and 52%) in a 15 ml Falcon tube. The solution was centrifuged at room temperature for 30 min at $1100 \times g$, resulting in three zones, peripheral blood mononuclear cells, PMNs, and red blood cells (RBCs), from the top to the bottom, respectively. PMNs were collected and centrifuged in 4 °C for 10 min at $350 \times g$. The pellet was collected and resuspended in 0.5 ml water for 25 s to lyse the remaining RBCs. Subsequently, 0.5 ml of 1.8% NaCl was added and the cells were centrifuged in 4 °C for 10 min at $350 \times g$, washed with 2 ml HEPES buffer, and centrifuged again in 4 °C for 10 min at $350 \times g$. Finally, the pellet was re-diluted in 1 ml DMEM complete. The viability of PMNs, checked with trypan blue dye, was over 98% and purity of PMNs (identified as CD11b+/Ly6G+ cells) was >90% by flow cytometry.

**FACS sorting and flow cytometry studies.** To obtain single lung cell suspensions, lungs were perfused with 20 ml PBS through the right ventricle, cut into small pieces, and digested for 1 h at 37 °C in 1 mg/ml collagenase A (Roche) and 0.05 mg/ml DNase I (Roche) in Hank's balanced salt solution. For flow cytometric analysis, single lung cells were stained with the following antibodies: anti-CD45-APC (Clone

30-F11, BioLegend, #103111, 1:200 dilution), anti-MHCII-FITC (Clone M5/114.15.2, BioLegend, #107605, 1:100 dilution), anti-F4/80-PE (Clone BM8, BioLegend, #123109, 1:100 dilution), anti-CD11c-PerCP-Cy5.5 (Clone N418, BioLegend, #117328, 1:100 dilution), anti-CD11b-PE-Cy7 (Clone M1/70, BioLegend, #101215, 1:200 dilution), anti-Ly6G-PE (Clone 1A8, BioLegend, #127607, 1:200 dilution), and anti-Ly6C-FITC (Clone HK1.4, BioLegend, #128005, 1:200 dilution). Flow cytometric data were collected on a MoFloT High-Performance Cell Sorter and were analyzed with FlowJo, version 8.7.1 (Treestar). AMs and IMs were sorted by flow cytometry based on their differential F4/80/CD11c/MHCII expression, as previously described[54]. PMNs were sorted as MHCII−/CD11b+/Ly6G+ cells. Isolated cells were cultured in RPMI-1640 medium supplemented with 10% fetal calf serum, 2 mM L-glutamine, 1 mM sodium pyruvate, 0.1 mM nonessential amino acids, 50 μM β-mercaptoethanol, 50 μg/ml streptomycin, and 50 IU/ml penicillin (all from Invitrogen), fixed and stained for confocal imaging. Flow cytometric data were collected on a MoFloT High-Performance Cell Sorter and were analyzed with FlowJo, version 8.7.1 (Treestar).

For FACS analysis in CD11c-DTR mice, the following antibodies were used: anti-CD45-APC-Cy7 (Clone 30-F11, BioLegend, #103116, 1:200), anti-CD11c-PerCP-Cy5.5 (Clone N418, BioLegend, #117328, 1:200), anti-I-A/I-E-PE-Cy7 (M5/114.15.2, BioLegend, #107630, 1:200), and anti-Singlec-F Alexa Fluor® 647 (E50-2440, BD Pharmingen, #562680, 1:200). Cells were acquired in a FACS Aria IIu (BD Biosciences) and data were analyzed with the FACSDIVA software (BD Biosciences).

**Histopathological and immunohistochemistry studies.** Lungs were fixed in 10% formalin, paraffin embedded, cut in 4-μm sections, and stained with hematoxylin and eosin. For immunohistochemistry studies, the anti-CD11c Abs (HL3; BD Biosciences, 1:200 dilution) and anti-CD68 (FA-11, 137001, BioLegend, 1:200 dilution) primary antibodies were used for detection of CD11c and CD68 in tissue. The slide-mounted sections were baked for 10 min at 60 °C, deparaffinized with two xylene washes, rehydrated through a series of graded alcohol washes, rinsed in water, and washed with 0.1 M PBS (pH 7.4) containing 0.01% Tween-20. Heat-induced antigen retrieval was performed in a steamer using target retrieval solution (Dako S1700). Endogenous peroxidase was blocked with 3% $H_2O_2$ for 10 min. The slides were then incubated in blocking solution (serum-free protein block, Dako X0909) for 20 min to block nonspecific binding. The primary antibodies were added to the slides and incubated overnight in a humidified chamber at 40 °C. Detection was accomplished using an Envision_Horseradish Peroxidase Kit (Dako K0679). Immunostaining was revealed using 3,3′-diaminobenzidine. The slides were lightly counterstained with hematoxylin, progressively dehydrated through graded alcohols and xylene, and finally covered with a coverslip after mounting in DPX mounting medium. Slides were examined under an Olympus light microscope that was equipped with a ×40 objective. In certain experiments fungal conidia were counterstained with periodic acid-Schiff (PAS).

**RNA isolation from Rhizopus-infected BMDMs.** BMDMs ($2 \times 10^6$ cells per condition) obtained following 6 days of differentiation of BM cells of 12-week-old female C57BL/6 mice were seeded in 12-well plates and left overnight at 37 °C in DMEM media. Next, BMDMs were washed 2× with culture media, infected at an MOI of 1:2 (macrophage:fungal conidia) with R. delemar (strain 99-880), and 1 h later washed five times to remove extracellular conidia. At the indicated time point of infection (1, 4, and 18 h), BMDMs were removed by scraping, centrifuged at $400 \times g$, and lysed with 450 μl of RLT buffer + β-mercaptoethanol using the RNeasy Plant Mini Kit (Qiagen). As a negative control, $1 \times 10^7$ R. delemar conidia were added to the tissue culture plates containing medium alone without host cells and processed in parallel. Another control included RNA extracted from uninfected BMDMs. Then, each sample was sonicated using a sonication probe on ice 20 × 1 s (set 40). Afterwards, isolation of RNAs was performed according to the manufacturer's instructions.

**RNA-seq analysis.** All RNA-seq libraries (strand-specific, paired-end) were prepared with the TruSeq RNA Sample Prep Kit (Illumina). The total RNA samples were subjected to poly (A) enrichment as part of the TruSeq protocol. One hundred and fifty nucleotide sequences were determined from both ends of each complementary DNA fragment using the HiSeq platform (Illumina) as per the manufacturer's protocol. Sequencing reads were annotated and aligned to the UCSC mouse reference genome (mm10, GRCm38.75) as well as the R. delemar (strain 99-880) genome using TopHat2[55]. The alignment files from TopHat2 were used to generate read counts for each gene and a statistical analysis of differential gene expression was performed using the DE-seq package from Bioconductor[56]. A gene was considered differentially expressed if the P value for differential expression was <0.05 and the absolute log (base 2) fold change, relative to single organism control, was ≥1.

**Human studies.** Approval for the collection of clinical information and tissue samples from the patient with mucormycosis was obtained and the Ethics Committee of the University Hospital of Heraklion, Crete, Greece (5159/2014). The patient provided written informed consent in accordance with the Declaration of Helsinki

**Statistical analysis.** The data were expressed as mean ± SEM. Statistical significance of differences was determined by two-sided nonparametric Mann–Whitney test and one-way analysis of variance with the indicated post hoc test for multiple comparisons ($P < 0.05$ was considered statistically significant). Survival analysis was performed by log-rank (Mantel–Cox) test. Analysis was performed using the GraphPad Prism software (version V). All in vitro experiments were performed at least in triplicate and replicated at least twice.

## Data availability

All the data that support the findings of this study are available from the corresponding author (G.C.) upon reasonable request. All of the raw sequencing data from this study will have been submitted to the NCBI SRA database (http://www.ncbi.nlm.nih.gov) under accession code PRJNA407788.

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

## Acknowledgements

We are grateful to Zacharenia Vlata from the IMBB FACS sorting facility for assistance with FACS analyses and to Teclegiorgis Gebremariam for helping with the animal studies. We are grateful to the EPR facilities available at the French research infrastructure IR-RPE CNRS 3443. I.K. is supported by the Onassis Foundation under the Special Grant and Support Program for Scholars Association Members (Grant No. R ZM 003-1/2016-2017; G.C. and A.M.A. are supported by a grant from the Greek State Scholarship Foundation (I.K.), G.C. is also supported by grants from the Hellenic General Secretariat for Research and Technology-Excellence program (ARISTEIA), and a Research Grant from Institute Merieux; A.S.I. is supported by Public Health Service grant from the National Institutes of Allergy and Immunology R01 AI063503 and U19AI110820. V.M.B. is supported by Public Health Service grant from the National Institutes of Allergy and Immunology U19AI110820.

## Author contributions

A.M.A. and I.K. performed and analyzed most of the experiments in this study and participated in their design. K.T. and E.A. performed experiments with CD11c-DTR mice and analyzed data. E.D. and M.T. performed histopathology studies. C.B. performed experiments with albino *Rhizopus* conidia. S.S.M.S. performed studies on physicochemical characterization of *Rhizopus* melanin. V.M.B., A.C.S., and C.M.C. performed dual RNA-seq. T.A. and P.I. analyzed data and provided assistance in establishing protocols for immunostaining. K.S. participated in EM studies. C.P. and H.A.P. provided information on the patient with mucormycosis. V.B. and E.E. performed EPR studies on melanins. A.B. analyzed data and performed melanin purification studies. D.P.K. and G.S. analyzed data and provided helpful suggestions. A.S.I. participated in the design and supervision of experiments, performed experiments, analyzed data, and provided key materials, as well as suggestions throughout the study. G.C. conceived and supervised the study, performed experiments, was involved in the design and evaluation of all experiments, and wrote the manuscript together with A.S.I. along with comments from co-authors.

## Additional information

**Competing interests:** The authors declare no competing interests.

