## [Peer Review File · Nature Communications]

Reviewers' comments:

Reviewer #1 (Remarks to the Author):

The manuscript Andrianaki and colleagues examines the lung immune response to *Rhizopus*, a devastating cause of mucormycosis in specific immune compromised patient populations. The study focuses primarily on macrophage - *Rhizopus* interactions and the finding that the fungus can induce persistence within macrophages, in part via phagolysosomal arrest via the action of melanin. The authors go on to examine the role of macrophages *in vivo* and the role of iron limitation as a host defense strategy within infected macrophages. The strength of the work is the level of detail and depth compared to prior studies on mucormycosis and the insight that this pathogen triggers fundamentally different immune responses compared to other filamentous molds (i.e., *Aspergillus fumigatus*).

The paper is well-written and the data are well-presented.

Major comments:

1. In figure 1, the distinction between phagocytosis and mere surface association/binding is not clear from the reported assays, as it relates to Figure 1G and 1H. Please explain.

2. While it is understandable that the authors used BMDMs for many *in vitro* studies in Fig. 2, 3, it would be informative to verify key phenotypes in alveolar macrophages, since BMDMs may not behave similarly to AMs with respect to Mucorales interactions (e.g., lack of intracellular swelling, and phagolysosomal arrest).

Minor comments:

Fig. 1A - would show on log scale and indicate how day 5 CFUs compare to the initial inoculum. Yes, *R. oryzae* "persists" compared to *A. fumigatus* but this "persister" fraction represents 2% of the inoculum. This point should be acknowledged and discussed. Another interpretation of the findings would be simply delayed clearance. To distinguish, it would be helpful to show additional data points before and after day 5.

p7. the term "prolonged intracellular lifestyle" is misleading given the 5 day time point examined and lack of evidence that *Rhizopus* can replicate or grow within this niche. The distinction here is to an organism like *Histoplasma* which can establish a replicative niche within macrophages.

When the authors infected mice with swollen conidia (Fig. 5), how much swelling did they observe? How much larger were swollen fungal cells than their resting counterparts?

Statistics:

The authors should use non-parametric tests throughout. Use of student t-test should be avoided because data are not parametric.

Reviewer #2 (Remarks to the Author):

This manuscript by Andrianaki and coworkers describes a host defense mechanism against Mucorales respiratory infection, in which iron restriction by macrophages leads to an arrest of Mucorales development with two main consequences : inhibition of phagosome maturation and persistence of the fungal pathogen within phagocytic cells.

Overall, this well-conducted study constitutes a considerable amount of work, and the findings represent an important step forward for a better comprehension of physiopathology of Mucorales infection.

Therefore, I consider this manuscript suitable for publication in Nature Communications after careful correction of minor errors and/or answering of following questions:

- the title can't be "...during respiratory fungal infection". The observations made are true only for Mucorales species (what is observed for *Rhizopus* is not seen for *Aspergillus*). I recommend to change the title to "... during respiratory mucormycosis"
- why some experiments are performed on *R. oryzae* while other are made with *R. delemar*? I may suppose that the results would be identical between the two species. Please explain
- why two different methods were used to determine the killing of fungal cells by macrophages (ie PI for *A. fumigatus* and subsequent germination for *R. oryzae*)? Moreover, the method used for determination of *R. oryzae* killing lacks a control, in which you verify that the incubation of *R. oryzae* spores that have never been exposed to macrophages for 4h at 37°C, in 5% CO₂ in DMEM complete medium leads to a 100% germ tube formation.
- regarding RNA extraction for RNAseq experiments, one may anticipate that RNA plant kit from Qiagen is not optimized for murine cells RNA extraction, and subsequent differential expression analysis could be biased by this technical limitation.
- rephrase 3rd sentence of the abstract as follows "[...] results in surface retention of melanin that induces phagosome maturation arrest through inhibition of LC3-associated phagocytosis."
- p5 : a part of the 2nd sentence is missing, did you mean "Ferroxamine is the iron-rich form of deferoxamine which is utilized by the fungus as a xenosiderophore."?
- p7 : "[...]in the lungs of immunocompetent mice (Figure 1E and 1F)." Delete "Figure 1E" between brackets since this panel only refers to total phagocytic cells count in the lungs, and not to *Rhizopus*-associated cells.
- p8 : please moderate the statement "[...], different clinical isolates of Mucorales were resistant to killing by BMDMs [...]", since on Fig. 2E, a killing of *R. oryzae* is seen.
- p8 : 2nd paragraph, 1st sentence: what is the link between persistence inside macrophages and innate antifungal resistance? Please develop or delete the second part of the sentence.
- p8 : Even if the concluding sentence of paragraph 2 remains true (persistence cannot be explained by resistance to oxidative stress), your results suggest that *R. oryzae* is more susceptible to H₂O₂ and on the opposite less susceptible to lysosomal extract as compared to *A. fumigatus*. Please modify the sentence accordingly ("both fungi displayed comparable degree of susceptibility")
- p9 : last sentence of 2nd paragraph : on Fig. 3C, Rab5+ macrophages are not "completely" absent, please correct
- p12 : rephrase the 2nd sentence of the last paragraph as follows "Interestingly, survival, histopathology and fungal loads experiments demonstrate that liposome depletion resulted in significant increase in susceptibility of mice to mucormycosis as compared to control liposomes (Figure 5D, 5E, 5F)."
- p15 : a part is missing in the 1st sentence "[...] we tested the ability of *Rhizopus* mutant defective in pathways of iron assimilation [...] and neutropenia." Which ability has been tested?
- p15 : rephrase 4th sentence as follows "Notably, the *Rhizopus fob1/2* mutant with defect in DFO uptake displayed selective impaired germination following [...]"

- p16 : extra comma after "by other filamentous fungi" in the 3rd sentence of the Discussion
- p16 : in reference to the remark concerning the title , change "Finally, we identify nutritional immunity [...] a major host defense mechanism during respiratory fungal infection." to "Finally, we identify nutritional immunity [...] a major host defense mechanism during respiratory Mucorales infection." and "[...] in future design of novel therapeutics against respiratory fungal diseases." to "[...] in future design of novel therapeutics against mucormycosis."
- p39 : "lungs" is misspelled in "Representative photomicrographs of the lungs", this typo is copy-pasted p43
- p41 : rephrase (D-E) as follows : "BMDMs were preloaded with FITC-Dextran, infected as in A, and phagolysosomal fusion [...]"
- p42 : "fluorescent" is misspelled in " (E, G) Representative fluorescent images [...]"
- Figure 2 : panels G and I give the same information, delete panel I; likewise, information given in panel K is summarized in panel J, delete panel K (NB : "necrotic" is misspelled in panel J)
- overall, the Materials and Methods section is poorly written as compared to the other parts of the manuscript. Please, proofread carefully this section, being particularly attentive to writing convention for units, italics for species name, grammar... Correct the following mistakes:
 - o p21 : 200 g dextrose per liter in YNB+CSM-URA, I think it is 20 g per liter
 - o p23 : "using (SPECTRO UV-VIS)." a part of the sentence is missing
 - o p24 : 1M HCL must be 1 M HCl; the sentence "The solvent system [...] was 5 min." must be re-written
 - o p25 : delete "phagosome" in "were assessed as described previously phagosome." (two first lines); for intratracheal administration of clodronate (which is misspelled) liposomes, give the quantity used and not the volume...
 - o p27 : a full stop is missing after "Adobe Photoshop CS2"; "according to established protocols in our lab." This explanation doesn't look very rigorous, please delete; "Protocol for lysosomal extracts and incubation with *R. oryzae* and *A. fumigatus*." is not a suitable title for this paragraph, change to "Lysosomal extracts preparation"; the sentence from "At least 3 x 10⁸ freshly collected BMDMs [...]" to "[...], resulting in crude lysosome extract." can be deleted if this protocol is identical to the procedure described in reference 53; "concentration" is misspelled in "in increasing concentration of 10%, 25 % and 50%"
 - o p28 : change the end of the first paragraph to "The absorbance of formazan, the XTT reduction product, was read at 450 and 655 nm on a Bio-Rad 680 microplate spectrophotometer. The percentage of metabolic activity was determined as follows : % metabolic activity = 100 x (OD₄₅₀-OD₆₅₅) experiment/ (OD₄₅₀-OD₆₅₅) control. Fungal killing was evaluated by plating on Sabouraud agar a 100-fold dilution of each well in sterile PBS."; change the second sentence of the second paragraph to "Briefly, BMDMs were collected, counted, and inoculated in DMEM-Glutamax, 10% FBS, 1% streptomycin in 6-well plates."; a space is missing between 1 mM and glycerophosphate in the acid phosphatase reaction buffer composition; delete "the" before acid phosphatase reaction buffer in the sentence "The cells were then rinsed three times with the acid phosphatase reaction buffer [...]"
 - o p29 : conidia should not be italicized in first sentence of second paragraph; "centrifuged at 1000 rpm for 10 min at 40°C", 40°C, sure?; change "*A. fumigatus* killing [...]" to "*Aspergillus fumigatus* killing [...]"; last sentence of second paragraph, change to "Killing of *R. oryzae* was assessed using a Bürker counting chamber, [...]"
 - o p30 : Murin PMNs isolation, 2nd sentence and so on "[...] was collected and flushed at room temperature [...]. The cells were centrifuged at room temperature for 10 min at 350 g and resuspended in 2 ml PBS/EDTA. The cells were carefully placed on top of 2 ml of three different

Percoll concentrations (75, 67 and 52%) in a 20 ml Falcon tube. The solution was centrifuged at room temperature [...], from top to bottom, respectively. [...], PMNs were collected and centrifuged at 4°C for 10 min at 350 g [...] and the cells were centrifuged at 4°C for 10 min at 350 g."

o p32 : change sentences 2 and 3 as follows : "[...] with R. delemar (strain 99-880) and 1 h later washed 5 times to [...] BMDMs were removed by scraping and centrifuged at 400 g [...]"

Reviewers' comments:

Reviewer #1 (Remarks to the Author):

General comments

- 1) ***The strength of the work is the level of detail and depth compared to prior studies on mucormycosis and the insight that this pathogen triggers fundamentally different immune responses compared to other filamentous molds (i.e., Aspergillus fumigatus). The paper is well-written and the data are well-presented.***

We thank the reviewer for the favorable comments.

Major comments:

- 1) ***In figure 1, the distinction between phagocytosis and mere surface association/binding is not clear from the reported assays, as it relates to Figure 1G and 1H. Please explain.***

Because flow cytometry of fluorescent labeled conidia of *Rhizopus* is not able to discriminate between surface binding and phagocytosis, we performed confocal imaging with serial z-sections of 0.3 mm step size across the cell surface of sorted phagocytes, which had been previously labeled with Cathepsin D to detect intracellular structures. In the revised manuscript we have included a new representative confocal image (**Fig. 1g**) showing cross section analysis in alveolar macrophages (AMs) that allows for definitive discrimination of intracellular conidia from conidia associated/bound to the cell surface of AMs. Both *x-y* and *z-x* projections of merged images of labeled *Rhizopus* conidia with intracellular membranes of AMs are shown (**embedded Image**).

2) While it is understandable that the authors used BMDMs for many *in vitro* studies in Fig. 2, 3, it would be informative to verify key phenotypes in alveolar macrophages, since BMDMs may not behave similarly to AMs with respect to Mucorales interactions (e.g., lack of intracellular swelling, and phagolysosomal arrest).

We have performed additional studies on AMs obtained with bronchoalveolar lavage from immunocompetent mice at different time points of intratracheal infection with conidia of either *A. fumigatus* or *R. oryzae*. In pilot studies we found that at 2h of infection over 90% of conidia of both fungi had been phagocytosed by AMs. Because LC3 recruitment is an early and transient event during phagosome biogenesis, we assessed LC3⁺ phagosome (LAPosome) formation at an early time point of phagosome formation (2h of infection) following infection of GFP-LC3 mice. In addition, we assessed phagolysosomal fusion by Cathepsin-D staining at 4h of infection of C57BL/6 (B6). In agreement with our findings in BMDMs, we found that as opposite to *A. fumigatus*, *R. oryzae* completely inhibited LAPosome formation and phagolysosomal fusion in AMs. We have included these new findings in **new Supplementary Fig. 5** of the revised manuscript.

Furthermore, comparative analysis of the size conidia revealed no significant evidence of intracellular swelling of *R. oryzae* conidia at 24h of infection as compared to 2h (early stages of phagocytosis). In contrast, there was a significant increase in conidial size (swelling) of

intracellular *A. fumigatus* conidia inside AMs at 24h. These results corroborate the findings of studies with *R. oryzae* and *A. fumigatus* conidia in BMDMs and have been included in **Supplementary Figure 6**.

Minor comments:

- 1) *Fig. 1A - would show on log scale and indicate how day 5 CFUs compare to the initial inoculum. Yes, R. oryzae "persists" compared to A. fumigatus but this "persister" fraction represents 2% of the inoculum. This point should be acknowledged and discussed. Another interpretation of the findings would be simply delayed clearance. To distinguish, it would be helpful to show additional data points before and after day 5.*

We performed additional experiments in immunocompetent mice infected with 5×10^6 conidia of either *A. fumigatus* or each of the two different *Rhizopus* clinical isolates that have been used throughout the studies (*R. oryzae*, or *R. delemar*). We characterized the kinetics of fungal load, expressed as \log_{10} CFU, at different time points of infection (2, 5, or 10 days) as compared to the initial inoculum (time point 0 h). We present these data as a **new Fig. 1a** of the revised manuscript. When compared to the initial inoculum recovered from the lungs ($\approx 1 \times 10^6$ CFUs/lungs, 0 h), we found evidence of “persistence” of $\approx 10\%$ and 1% of conidia of both *Rhizopus* clinical isolates on Day 5 and Day 10 post infection, respectively. In contrast, there was evidence of fungal clearance on Day 5 of infection of *A. fumigatus*. Collectively, we believe that these studies justify the use of term “**intracellular persistence**” of *Rhizopus* conidia, which has been used throughout the manuscript. Instead, we have removed the term “intracellular lifecycle”.

- 2) *p7. the term "prolonged intracellular lifestyle" is misleading given the 5 day time point examined and lack of evidence that Rhizopus can replicate or grow within this niche [sic]. The distinction here is to an organism like Histoplasma which can establish a replicative niche within macrophages.*

We appreciate the reviewer’s comment. Indeed, we have no evidence of intracellular proliferation of *Rhizopus* conidia inside macrophages. Therefore, we have removed the phrase “prolonged intracellular lifecycle”. On the other side, we clearly observe “persistent” conidia of *Rhizopus* inside macrophages of immunocompetent mice and confirm this observation in a patient with mucormycosis. Intracellular persistence has been recently shown as an important virulence strategy of other typical extracellular pathogens, including *Staphylococcus aureus* (ref. 42) and *Streptococcus pneumoniae* (Ercoli G et al., **Nat Microbiol.** 2018 May;3(5):600-610), which accounts for treatment failures (Kim W et al., **Nature.** 2018 Apr 5;556(7699):103-107) and relapse of the infection. **Therefore, in view of the important pathogenetic role of intracellular persistence in mucormycosis, we prefer to use the term “intracellular persistence” instead of “delayed clearance”.**

- 3) *When the authors infected mice with swollen conidia (Fig. 5), how much swelling did they observe? How much larger were swollen fungal cells than their resting counterparts?***

We have performed additional histopathology studies of immunocompetent mice during the course of infection with either swollen or resting conidia of *Rhizopus* (0h, 24h) and present the findings in Supplementary Figure 12. Notably, we found no evidence of germling formation upon infection of mice with swollen conidia (4 h incubation in culture media). As expected, at initial time point of infection (0 h), swollen conidia were larger in size than their resting counterparts. At 24 h of infection, a significant proportion of dormant conidia resided inside macrophages, whereas there was evidence of invasive fungal growth with hyphae formation of swollen conidia without evidence of phagocytosis. Importantly, the mechanisms of inhibition of phagocytosis of swollen conidia by AMs and the induction of acute lethality within few days of infection of immunocompetent mice might not be directly related to the differences in conidial size between dormant and swollen conidia of *Rhizopus*. Ongoing work in our laboratories will address this important pathogenetic mechanism.

- 4) *Statistics: The authors should use non-parametric tests throughout. Use of student t-test should be avoided because data are not parametric.***

We have performed statistical comparisons with the use of non-parametric tests throughout the study as per the reviewer's suggestions and include these data in the revised manuscript figure legends and text.

Reviewer #2 (Remarks to the Author):

General comments

Overall, this well-conducted study constitutes a considerable amount of work, and the findings represent an important step forward for a better comprehension of physiopathology of Mucorales infection. Therefore, I consider this manuscript suitable for publication in Nature Communications after careful correction of minor errors and/or answering of following questions:

We thank the reviewer for the favorable comments.

- 1) *the title can't be "...during respiratory fungal infection". The observations made are true only for Mucorales species (what is observed for Rhizopus is not seen for Aspergillus). I recommend to change the title to "... during respiratory mucormycosis"***

We appreciate the reviewer's comment. In the revised manuscript we have changed the title according to the reviewer suggestion to "...during pulmonary mucormycosis".

- 2) ***why some experiments are performed on *R. oryzae* while other are made with *R. delemar*? I may suppose that the results would be identical between the two species. Please explain***

We have performed experiments with two clinical isolates of *Rhizopus* to increase the physiological relevance of our findings. Both *Rhizopus* species induced the same phenotype in terms of in vivo “persistence” in the lungs of immunocompetent mice (**Fig 1a, 1b**), phagosome biogenesis blockade and killing by BMDMs (**Figure 2, Supplementary Fig. 3a**). Importantly, *R. delemar* (strain 99-880) is a reference isolate that has been extensively characterized in virulence studies and the first *Rhizopus* strain with annotated genome. Therefore, we have extensively validated the response of macrophages to both *Rhizopus* clinical isolates (*R. oryzae* and *R. delemar*) and subsequently used the reference strain *R. delemar* for transcriptomics.

- 3) ***why two different methods were used to determine the killing of fungal cells by macrophages (ie PI for *A. fumigatus* and subsequent germination for *R. oryzae*)? Moreover, the method used for determination of *R. oryzae* killing lacks a control, in which you verify that the incubation of *R. oryzae* spores that have never been exposed to macrophages for 4h at 37°C, in 5% CO2 in DMEM complete medium leads to a 100% germ tube formation.***

In our previous work we have optimized a reliable and simple protocol for the assessment of killing of *A. fumigatus* conidia by monocytes/macrophages with the use of PI staining, a vital dye that is extensively used to discriminate live and dead cells.

Unfortunately, PI staining of *Rhizopus* conidia resulted in unspecific fluorescence of both live and dead fungal cells. Therefore, we assessed BMDMs-mediated killing of *Rhizopus* conidia by direct evaluation of germination rate of intracellular conidia following BMDMs lysis at different time points of infection. Importantly, the percentage of killing of *Rhizopus* conidia by BMDMs was always normalized to the germination of control *R. oryzae* conidia that were sonicated for 5 sec (to account for the effect of cell lysis on viability) in the absence of BMDMs. Germination of control conidia (without macrophages) sonicated for 5 sec in culture media was always > 95 %. We have clarified this point in the methods section of our revised manuscript. Finally for consistency, we have also analyzed the killing of *A. fumigatus* conidia by BMDMs assessed based on germination rate of conidia (**Supplementary Fig. 3**)

- 4) ***regarding RNA extraction for RNAseq experiments, one may anticipate that RNA plant kit from Qiagen is not optimized for murine cells RNA extraction, and subsequent differential expression analysis could be biased by this technical limitation.***

The protocol of simultaneous RNA extraction from fungal and mammalian cells has been previously optimized and successfully used in published transcriptomic studies of our group in Nature Communications (Chibucos MC et al., Nat Commun. 2016 Jul 22;7:12218. doi: 10.1038/ncomms12218.). In addition, the extracted RNA passed successfully all quality control

tests before transcriptomic analysis. Therefore, we do not anticipate a significant effect of this method of RNA extraction in transcriptomic analysis.

- 5) *rephrase 3rd sentence of the abstract as follows "[...] results in surface retention of melanin that induces phagosome maturation arrest through inhibition of LC3-associated phagocytosis."*

We have rephrased the abstract accordingly.

- 6) *p5: a part of the 2nd sentence is missing, did you mean "Ferroxiamine is the iron-rich form of deferoxamine which is utilized by the fungus as a xenosiderophore."?*

Yes, and we have corrected the typo.

- 7) *p7: "[...]in the lungs of immunocompetent mice (Figure 1E and 1F)." Delete "Figure 1E" between brackets since this panel only refers to total phagocytic cells count in the lungs, and not to Rhizopus-associated cells.*

We have removed the reference to Figure 1E in the sentence.

- 8) *p8: please moderate the statement "[...], different clinical isolates of Mucorales were resistant to killing by BMDMs [...]", since on Fig. 2E, a killing of R. oryzae is seen.*

We have performed killing studies of different clinical isolated of *Rhizopus* (*R. oryzae*, and *R. delemar*) by BMDMs, and present these studies in Fig. 2E and Supplementary Figure 3. We have rephrased the statement to “different clinical isolates of *Rhizopus*”.

- 9) *p8: 2nd paragraph, 1st sentence: what is the link between persistence inside macrophages and innate antifungal resistance? Please develop or delete the second part of the sentence.*

We agree with the reviewer on the lack of a direct link on resistance to killing by macrophages and inherent resistance to antifungal agents. Therefore, we deleted the second part of this sentence.

- 10) *p8: Even if the concluding sentence of paragraph 2 remains true (persistence cannot be explained by resistance to oxidative stress), your results suggest that R. oryzae is more susceptible to H2O2 and on the opposite less susceptible to lysosomal extract as compared to A. fumigatus. Please modify the sentence accordingly ("both fungi displayed comparable degree of susceptibility")*

We have modified the sentence accordingly.

- 11) *p9: last sentence of 2nd paragraph : on Fig. 3C, Rab5+ macrophages are not "completely" absent, please correct*

We have corrected as follow “while there was no evidence of Rab5 localization in *R. oryzae*-containing phagosomes”.

- 12) *p12 : rephrase the 2nd sentence of the last paragraph as follows "Interestingly, survival, histopathology and fungal loads experiments demonstrate that liposome depletion resulted in significant increase in susceptibility of mice to mucormycosis as compared to control liposomes (Figure 5D, 5E, 5F)."*

We have rephrased accordingly.

- 13) *p15: a part is missing in the 1st sentence "[...] we tested the ability of Rhizopus mutant defective in pathways of iron assimilation [...] and neutropenia." Which ability has been tested?*

We added the missing sentence “to germinate intracellularly following iron supplementation”.

- 14) *p15: rephrase 4th sentence as follows "Notably, the Rhizopus job1/2 mutant with defect in DFO uptake displayed selective impaired germination following [...]"*

We have rephrased the sentence accordingly.

- 15) *p16 : extra comma after "by other filamentous fungi" in the 3rd sentence of the Discussion*

Extra comma has been added.

- 16) *p16: in reference to the remark concerning the title , change "Finally, we identify nutritional immunity [...] a major host defense mechanism during respiratory fungal infection." to "Finally, we identify nutritional immunity [...] a major host defense mechanism during respiratory Mucorales infection." and "[...] in future design of novel therapeutics against respiratory fungal diseases." to "[...] in future design of novel therapeutics against mucormycosis."*

We changed accordingly to “pulmonary mucormycosis” and “mucormycosis”.

17) **p39: "lungs" is misspelled in "Representative photomicrographs of the lungs", this typo is copy-pasted p43**

We have corrected the typo.

18) **p41: rephrase (D-E) as follows: "BMDMs were preloaded with FITC-Dextran, infected as in A, and phagolysosomal fusion [...]"**

We have rephrased accordingly.

19) **p42: "fluorescent" is misspelled in " (E, G) Representative fluorescent images [...]"**

We have corrected the typo.

20) **Figure 2: panels G and I give the same information, delete panel I; likewise, information given in panel K is summarized in panel J, delete panel K (NB : "necrotic" is misspelled in panel J)**

We have deleted panels I and K from Figure 2 and corrected the word necrotic in panel J (I in the revised Figure).

21) **overall, the Materials and Methods section is poorly written as compared to the other parts of the manuscript. Please, proofread carefully this section, being particularly attentive to writing convention for units, italics for species name, grammar... Correct the following mistakes:**

We have proofread the manuscript and made all appropriate corrections of the detected typos.

22) **p21 : 200 g dextrose per liter in YNB+CSM-URA, I think it is 20 g per liter**

The typo has been corrected.

23) **p23: "using (SPECTRO UV-VIS)." a part of the sentence is missing**

We have corrected the missing part of the sentence "with the use of a SPECTRO UV-VIS spectrophotometer".

24) **p24 : 1M HCL must be 1 M HCl; the sentence "The solvent system [...] was 5 min." must be re-written**

We have completely rephrased this part of the methods.

25)p25: delete "phagosome" in "were assessed as described previously phagosome." (two first lines); for intratracheal administration of clodronate (which is misspelled) liposomes, give the quantity used and not the volume

We corrected the typos, deleted the word phagosome and added information on the amount of clodronate used in quantity.

26)p27: a full stop is missing after "Adobe Photoshop CS2"; "according to established protocols in our lab." This explanation doesn't look very rigorous, please delete; "Protocol for lysosomal extracts and incubation with *R. oryzae* and *A. fumigatus*." is not a suitable title for this paragraph, change to "Lysosomal extracts preparation"; the sentence from "At least 3 x 10⁸ freshly collected BMDMs [...]" to "[...], resulting in crude lysosome extract." can be deleted if this protocol is identical to the procedure described in reference 53; "concentration" is misspelled in "in increasing concentration of 10%, 25 % and 50%"

All the changes and corrections have been made.

27)p28: change the end of the first paragraph to "The absorbance of formazan, the XTT reduction product, was read at 450 and 655 nm on a Bio-Rad 680 microplate spectrophotometer. The percentage of metabolic activity was determined as follows : % metabolic activity = 100 x (OD₄₅₀-OD₆₅₅) experiment/ (OD₄₅₀-OD₆₅₅) control. Fungal killing was evaluated by plating on Sabouraud agar a 100-fold dilution of each well in sterile PBS."; change the second sentence of the second paragraph to "Briefly, BMDMs were collected, counted, and inoculated in DMEM-Glutamax, 10% FBS, 1% streptomycin in 6-well plates."; a space is missing between 1 mM and glycerophosphate in the acid phosphatase reaction buffer composition; delete "the" before acid phosphatase reaction buffer in the sentence "The cells were then rinsed three times with the acid phosphatase reaction buffer [...]"

We have performed all the changes and corrections.

28)p29 : conidia should not be italicized in first sentence of second paragraph; "centrifuged at 1000 rpm for 10 min at 40°C", 40°C, sure?; change "A. fumigatus killing [...]" to "Aspergillus fumigatus killing [...]"; last sentence of second paragraph, change to "Killing of *R. oryzae* was assessed using a Bürker counting chamber, [...]"

All requested changes have been made.

29)p30: Murin PMNs isolation, 2nd sentence and so on "[...] was collected and flushed at room temperature [...]. The cells were centrifuged at room temperature for 10 min at 350 g and resuspended in 2 ml PBS/EDTA. The cells were carefully placed on top of 2 ml of three different Percoll concentrations (75, 67 and 52%) in a 20 ml Falcon tube. The solution was centrifuged at room temperature [...], from top to bottom, respectively.

[...],PMNs were collected and centrifuged at 4°C for 10 min at 350 g [...] and the cells were centrifuged at 4°C for 10 min at 350 g."

All the suggested changes have been made.

30)p32 : *change sentences 2 and 3 as follows : "[...] with R. delemar (strain 99-880) and 1 h later washed 5 times to [...] BMDMs were removed by scraping and centrifuged at 400 g [...]"*

We have made the changes.

REVIEWERS' COMMENTS:

Reviewer #1 (Remarks to the Author):

The authors have adequately responded to my concerns about the manuscript.

Reviewer #2 (Remarks to the Author):

The reviewer's comments have been correctly addressed, this manuscript is now suitable for publication.

**Iron restriction inside macrophages regulates pulmonary host defense against
Rhizopus species**

July 20, 2018

REVIEWERS' COMMENTS: Reviewer #1 (Remarks to the Author): The authors have adequately responded to my concerns about the manuscript.

Reviewer #2 (Remarks to the Author): The reviewer's comments have been correctly addressed, this manuscript is now suitable for publication.

Reply: We thank both reviewers for their positive response and for considering our revised manuscript suitable for publication in Nature Communications.

Sincerely,

Georgios Chamilos, MD
Department of Medicine,
University of Crete, Unit 3D72,
Heraklion, Crete, Greece, 71110,
Telephone: +30-(2810)-394560
Fax: +30-(2810)-394626
E-mail: hamilos@imbb.forth.gr

Ashraf S. Ibrahim, PhD
Los Angeles Biomedical Research Institute,
Division of Infectious Diseases,
Harbor-UCLA Medical Center,
1124 West Carson Street,
St. John's Cardiovascular Research Center,
Torrance, California 90502, USA

Phone: 310.222.6424

E-mail: ibrahim@labiomed.org